# Integrated water vapor and liquid water path retrieval using a single-channel radiometer

Anne-Claire Billault-Roux[1] and Alexis Berne[1]

[1]Environmental Remote Sensing Laboratory, Swiss Federal Institute of Technology, Lausanne

**Correspondence:** Alexis Berne (alexis.berne@epfl.ch)

**Abstract.** Microwave radiometers are widely used for the retrieval of Liquid Water Path (LWP) and Integrated Water Vapor (IWV) in the context of cloud and precipitation studies. This paper presents a new site-independent retrieval algorithm for LWP and IWV, relying on a single-frequency 89-GHz ground-based radiometer. A statistical approach is used, based on a neural network, which is trained and tested on a synthetic data set constructed from radiosonde profiles worldwide. In addition to 89-GHz brightness temperature, the input features include surface measurements of temperature, pressure and humidity, as well as geographical information and, when available, estimates of IWV and LWP from reanalysis data. An analysis of the algorithm is presented to assess its accuracy, the impact of the various input features, as well as its sensitivity to radiometer calibration and its stability across geographical locations. While 89-GHz brightness temperature is crucial to LWP retrieval, only moderately does it contribute to IWV estimation, which is more constrained by the additional input features. The algorithm is shown to be quite robust although its accuracy is inevitably lower than that obtained with state-of-the-art multi-channel radiometers, with a relative error of 18 % for LWP (on cloudy cases with LWP > 30 $\mathrm{g\,m^{-2}}$) and 6.5 % on IWV. The highest accuracy is obtained in mid-latitude environments with a moderately moist climate, which are more represented in the training dataset. The new method is then implemented and evaluated on real data that were collected during a field deployment in Switzerland and during the ICE-POP 2018 campaign in South Korea.

## 1 Introduction

Clouds play a key, though complex, role in the atmosphere's radiative balance and global circulation (Hartmann and Short, 1980; Slingo, 1990; Hartmann et al., 1992; Wang and Rossow, 1998; Stephens, 2005; Mace et al., 2006; McFarlane et al., 2008), and cloud studies have thus been propelled to the forefront of climate research. One of the core challenges is the monitoring, quantification and modeling of cloud liquid water, which has a significant contribution to radiative processes on a global scale. In this perspective, highly accurate methods were developed to retrieve liquid water path (LWP) as well as integrated water vapor (IWV) from microwave radiometer measurements, relying on the fact that water in its liquid and vapor phases is the main atmospheric contributor to brightness temperatures in millimeter wavelengths, outside of the oxygen window. On a different note, quantifying cloud liquid water content is also relevant to the field of snowfall studies. Identifying the presence of supercooled liquid water during a snowfall event is of paramount importance to the understanding of snowfall microphysics, for it drives riming of snow particles, which in turn affects the efficiency and the spatial distribution of precipitation (Saleeby

et al., 2011), as well as wet deposition of aerosols (Poulida et al., 1998). Improving the monitoring of cloud liquid water processes is thus valuable to climatological, meteorological and hydrological applications.

The quantitative retrieval of LWP from ground-based or satellite measurements of brightness temperature ($T_B$) at a single millimeter wavelength is an underdetermined problem. This brightness temperature results from the radiative contribution of gases and hydrometeors across the atmospheric column, and depends on the vertical profile of temperature. To lift this underdetermination, state-of-the-art retrievals of LWP and IWV rely on multi-frequency radiometers, which provide $T_B$ measurements in several microwave channels. This allows to separate the contributions of water vapor and liquid water (e.g. Westwater et al. (2001)) and, to some extent, retrieve the full profile of liquid water content and humidity in the atmospheric column (Löhnert et al., 2004). It should be noted that IWV retrievals with similar accuracy are obtained using GPS sensors, as first proposed by (Bevis et al., 1994), but this widely used technique does however not allow for joint retrieval of LWP.

Multi-frequency instruments are however not always available. It was shown (Küchler et al., 2017) that a radiometer channel at 89 GHz could be added to a W-band cloud radar operating at 94 GHz, thus allowing to have collocated measurements of radar variables and brightness temperature, and paving the way for an improved understanding of cloud and precipitation physics. Küchler et al. (2017) proposed a method to derive LWP estimates from the single-frequency measurements of brightness temperature, and the present study builds on those findings.

Two approaches are commonly considered for the retrieval of LWP and IWV from microwave radiometer measurements, as described in Turner et al. (2007) and Cadeddu et al. (2013). The first method relies on the reconstruction of atmospheric profiles, with a physical model that is iterated until modeled $T_B$s match the measured ones. Although this method is formally the most accurate (Turner et al., 2007), it requires more than one radiometer frequency to lift the problem's fundamental underdetermination, and is thus not applicable for this study. The other way to tackle the problem is to derive statistical relationships between $T_B$s and LWP and/or IWV based on synthetic datasets. This approach has been widely used, both for ground-based and satellite applications, with varying degrees of complexity in the algorithms (linear, quadratic, log fitting or using neural network architectures) (Karstens et al., 1994; Löhnert and Crewell, 2003; Mallet et al., 2002; Cadeddu et al., 2009). The retrieval coefficients that are computed with this method are usually site-specific, since they incorporate during the learning or regression stage the climatological features at the location of the dataset. The geographical range within which a site-specific algorithm could be reliable is difficult to estimate, especially if the orography of the region is complex, as highlighted by Massaro et al. (2015). In general, implementing a site-specific algorithm in a location with a different climatology is likely to yield erroneous retrievals (Gaussiat et al., 2007). In order to implement such an algorithm at another site, a new parameterization should be performed using a suitable dataset; but there might not always be enough reliable data available for this purpose. In order to avoid this lengthy process, and in the case of instruments that are intended to be deployed in various locations, a site-independent algorithm is more adequate (Liljegren et al., 2001).

The purpose of this study is to present a new site-independent method for the retrieval of both LWP and IWV, that relies on a single radiometer frequency. The regression is performed through a neural network, whose input consists of brightness temperature at 89 GHz, as well as surface measurements and geographical information. Those additional input features are

shown to be especially key to the retrieval of IWV. Although this new method comes with a loss of precision in comparison with state-of-the-art multi-frequency retrievals, its advantage is to be applicable in any location with a constrained uncertainty.

The following section describes the data used in the different steps of this study, from the design steps to the validation of the new method. Section 3 outlines the forward model that is used to build the synthetic dataset on which the LWP and IWV retrieval algorithms are trained. In Sect. 4, the design of the algorithms is detailed, and the results on the synthetic dataset are reviewed and analyzed in Sect. 5. An independent validation of the method is presented in Sect. 6 using two contrasted datasets that were collected during field deployments in Payerne (Switzerland) and in the Taebaek mountains (South Korea). A summary and conclusions are provided in Section 7.

## 2 Data

The present work is based on two types of data: a multiyear collection of radiosonde observations across the world (for training and testing of the retrieval algorithms) as well as sets of measurements from an 89-GHz radiometer deployed in various regions during field campaigns limited in time. Those two types of data are described below.

### 2.1 Radiosonde dataset

The design of a statistical algorithm requires a large dataset on which to perform statistical learning. Here, this dataset was built using radiosonde profiles collected in over 180 stations throughout the world, available through the University of Wyoming portal (Oolman, 2020). In total, $\sim 10^6$ radiosonde profiles are used, from 20 years of data (2000-2019). It was ensured that the data included radiosonde stations from all climatic regions and covering a wide range of altitudes (0 to 4000 m). However, lack of available data in some areas inevitably results in an unbalanced dataset, where polar and tropical areas are under-represented compared to mid-latitudes, especially Europe. The possible impact on the performance of the algorithm is further discussed in Sect. 5.

A quality check was performed on each of the relevant variables (pressure, temperature, relative humidity), through the following steps: first, the minimum and maximum P (resp. T, RH) in a given range of altitudes were extracted from each radiosonde. When examining the distributions that are obtained, outliers were visible, which were then removed with a $10^{-4}$ quantile (upper and lower quantile). The atmospheric column was split into 9 ranges of altitudes, and this routine was performed for each. In total, 6395 profiles were flagged out and removed. It was ensured that this did not result in the systematic removal of some geographical locations. Following this step, the vertical profiles of pressure, temperature and relative humidity are used as input to the forward model, described in Sect. 3. The vertical extent of the atmospheric profiles ranges from 1 to 50 km, with a 0.25 quantile of 11km, meaning the profiles largely cover the lower troposphere. The vertical resolution is relatively low (0.37 km on average).

## 2.2 Field deployments

In the validation stage of this work, the new method was implemented using real 89-GHz radiometer data, that were collected during campaigns described below.

### 2.2.1 Instrument

The main instrument that was used for the implementation of the algorithm is the one described in Küchler et al. (2017), which is here referred to as WProf. This radar-radiometer system, conceived and built by RPG, consists of a 94-GHz Frequency-Modulated Continuous Wave (FMCW) cloud radar with an 89-GHz radiometer channel, which allows for joint active and passive retrievals of cloud and precipitation. In the data presented here, WProf was deployed together with a weather station that provided surface measurements of temperature, pressure and relative humidity.

### 2.2.2 Payerne 2017

The first data set on which the new algorithm was evaluated was collected during a field deployment in Payerne (Switzerland), at 450 m of altitude, in late spring 2017 (May 15th – June 15th). As a means of comparison, data from the Swiss meteorological institute (MeteoSwiss) was used. MeteoSwiss's facilities in Payerne comprise a multi-frequency radiometer with tipping-curve calibration, HATPRO (Rose et al., 2005; Löhnert and Maier, 2012). This state-of-the-art instrument retrieves LWP and IWV with a nominal accuracy of respectively $20 \mathrm{~g~m}^{-2}$ and $0.2 \mathrm{~kg~m}^{-2}$ (RPG Radiometer Physics GmbH, 2014). During this deployment, both WProf and HATPRO measured brightness temperatures with a high temporal resolution of the order of a few seconds. The instruments were located approximately 65 m apart; this distance is small enough that it should in general not affect the comparison of the retrieved values from the two instruments. However, in some rare cases, it is possible that a cloud would overpass one of the radiometers, but not the other, leading to a discrepancy in the measured brightness temperatures.

In addition, radiosondes are launched twice daily in Payerne by MeteoSwiss, allowing for the direct computation of IWV values, which are used as a further source of validation for the IWV retrieval algorithm.

### 2.2.3 ICE-POP 2018

The second dataset on which the new algorithm was tested was gathered during the ICE-POP 2018 campaign, which took place in South Korea during the 2017-2018 winter, in the context of the 2018 Olympic and Paralympic winter games in Pyeong Chang. A description of the data is presented in Gehring et al. (2020). During this campaign, the weather was generally cold and dry; nine precipitation events were recorded, and occasional fog was present (about 25 occurrences during the campaign timeframe). WProf was deployed from November 2017 to April 2018 in Mayhills, 50 km south-east of Pyeong Chang, at 789 m of altitude. This allows for an implementation of the algorithm in a different context than Payerne: i.e. in winter conditions and in a fully different geographical setting, located at a lower latitude and closer to the sea.

In this case, unlike in Payerne, no independent measurements of LWP are available; however, radiosondes were launched every 3 hours, thus providing a means of comparison for IWV retrievals, although only with a lower temporal resolution.

## 3 Forward model

In order to develop a statistical algorithm, a large amount of data is required to reliably perform the statistical learning phase. For this purpose, a synthetic data set was built, using as a starting point the radiosonde profiles described in the previous section. A two-step forward model was implemented, first to identify clouds in each profile and derive the corresponding liquid water content, then to compute the resulting 89-GHz brightness temperature. The different steps of this forward model are illustrated in the flowchart in Fig. 1.

### 3.1 Cloud liquid model

To derive profiles of liquid water content (LWC) from radiosonde profiles of atmospheric variables, the cloud model from Salonen and Uppala (1991) was used. Cloud boundaries are identified using a threshold $U_c$ on relative humidity, this threshold being pressure- and temperature-dependent, according to Eq. 1.

$$U_c = 1 - \alpha\sigma(1-\sigma)[1 + \beta(\sigma - 0.5)] \tag{1}$$

Here, $\sigma = \frac{P}{P_0}$ with $P$ and $P_0$ denoting respectively atmospheric pressure at the current level and at the ground. Corrections from Mattioli et al. (2009) are used for the coefficients $\alpha$ and $\beta$ of the Salonen model. Within the cloud layers, the liquid water profile is then calculated as a function of temperature and height above cloud base, following Eq. 2.

$$LWC = w_0 (\frac{h - h_b}{h_r})^a f(T) \tag{2}$$

where $f(T) = 1 + cT$ for $T \geq 0$ and $f(T) = exp(cT)$ for $T < 0$, with $T$ in °C, $a = 1.4$, $c = 0.04°\text{C}^{-1}$, $w_0 = 0.17\,\text{g m}^{-3}$, $h_r = 1.5\,\text{km}$, $h$ and $h_b$ denoting height and height of cloud base. There are some limitations to assuming a single universal cloud model, since it may fail to capture specific cloud properties in certain environments: more sophisticated and accurate models could be defined on a local geographical scale to counter this (e.g. Pierdicca et al. (2006)). However, given the stated objective of this study to design a non-site specific algorithm, it was considered preferable to assume a single universal liquid cloud model, in spite of its potential drawbacks.

A further limitation of the cloud model is related to the relatively low resolution of the atmospheric profiles extracted from the radiosonde data (c.f. Sect. 2.1) that are used as an input. This might result in a misrepresentation of the cloud layers in their detection and their size. In order to ensure that this forward model generated the least possible bias, its results were compared against LWP values from ERA5 reanalysis data (Copernicus Climate Change Service, 2020). Even though the model might fail, on a given occurrence, to reproduce the actual liquid water profile in the atmospheric column, it should not produce a significant bias on average. This condition guarantees that the synthetic dataset that is used for training contains realistic – if not real – profiles, and this should therefore not degrade the quality of the retrieval algorithm. This cloud model was chosen over other commonly used ones (Decker model, Salonen model without correction, c.f. Mattioli et al. (2009)) for it was found to produce the least bias when compared to ERA5 LWP values (mean bias of $14\,\text{g m}^{-2}$ vs. $26\,\text{g m}^{-2}$ (resp. $-24\,\text{g m}^{-2}$) for the unadjusted Salonen model (resp. the Decker model with 95 % threshold). Inevitably, when using this criterion for the choice

of the cloud liquid model, it is assumed that reanalysis values of LWP are themselves bias-free, which could be questioned, especially in extreme environments (e.g. Lenaerts et al., 2017).

## 3.2 Radiative transfer model

Ground-level brightness temperatures ($T_B$) at 89 GHz are simulated for each profile using the Passive and Active Microwave
TRansfer Model (PAMTRA (Maahn, 2015; Mech et al., 2020)) available at https://github.com/igmk/pamtra (last access: Nov 18th, 2020). As input to the radiative transfer calculations, vertical profiles of temperature, pressure, hydrometeor mixing ratio and water vapor mixing ratio are used. Gaseous absorption is calculated using the default parameters in PAMTRA, i.e. with the model proposed by Rosenkranz (1998) and modifications from Liljegren et al. (2005) and Turner et al. (2009). Liquid water absorption is modeled according to Ellison (2007). It should be kept in mind that some irreducible uncertainty remains tied to
the choice of these parameters in the radiative transfer model.

The cloud droplet size distribution (DSD) is chosen as a monodisperse distribution with radius $r_c = 20$ $\mu$m following Cadeddu et al. (2017), and scattering calculations are performed with Mie equations, assuming spherical particles. Let us note here that the exact choice of the DSD has little impact on $T_B$ modeling as long as the droplets are in the Rayleigh regime for the given frequency, since the emission cross-section in this regime is quasi-linearly related to the particle's volume. When the droplet
size deviates from this regime, for instance as droplets grow larger near the onset of precipitation, then the Rayleigh assumption falls short and higher-order terms in the Mie equations become non-negligible, which alters the modeling of $T_B$ (e.g., Zhang et al., 1999). This implies that the algorithm will output biased results when applied to raining cases, and should not be trusted in those circumstances. This shall be considered as an intrinsic limitation to the algorithm.

There is no clear-cut relation between LWP values and the occurrence of precipitation, although the general trend is that higher
LWP is related to more likely rain: as such, deviation from the Rayleigh regime is likely in high-LWP cases. In order to have a more rigorous grasp on when and how this drawback might affect the retrieval, criteria from Karstens et al. (1994) were used. In their study, the authors distinguished three types of liquid water clouds based on the value of LWC at a given altitude; for each category of cloud, a different characteristic radius is chosen for the DSD. Mie effects can start to become an issue in the second category of clouds (*cumulus congestus*), identified for LWC > 0.2 $\mathrm{g\,m^{-2}}$; in our dataset, the atmospheric profiles
where this LWC theshold is exceeded in at least one range gate have, on average, a total LWP $\geq 830$ $\mathrm{g\,m^{-2}}$, and around 2% of the entire dataset fall into this category. Taking the third category (*cumulonimbus*) with LWC > 0.4 $\mathrm{g\,m^{-2}}$, this applies to 1% of the entire dataset and the average LWP threshold increases to 1400 $\mathrm{g\,m^{-2}}$. Those values can serve as a benchmark to identify LWP values where Mie effects can typically contaminate the retrieval. However, edge cases can also exist where the total LWP is quite low, but a small layer of nearly-precipitating or drizzling cloud still contaminates the retrieval, without
featuring extremely high total LWP.

Finally, the forward model that is presented here does not include the contribution of ice clouds and snowfall. While radiative emission from ice and snow particles has a minor influence on brightness temperature when compared to emission from liquid droplets and water vapor, and is in general negligible, solid hydrometeors do contribute to microwave brightness temperature through the backscattering of surface radiation. Scattering from snowfall particles is difficult to model accurately, but Kneifel

et al. (2010) suggest that this effect could be notable during snowfall, in a way that is highly dependent on the microphysical properties of snowfall particles, and that would increase with their size. The present study does not take into account this process and could therefore yield biased results during intense snowfall events.

## 4 Design of the IWV and LWP retrieval algorithms

### 4.1 Input features

When a single frequency is available for the measurement of $T_B$, the problem's underdetermination can be partially relieved by including other available information in the retrieval's measurement vector. Adding further information allows to disentangle IWV and LWP, which could not be achieved from the sole measurement of 89-GHz $T_B$. In this study, several categories of variables were included in the input features. The first category consists of $T_B$ and higher order polynomials (up to fourth degree) and is expected to have the greatest importance in the retrieval of LWP, while the other categories would likely be more correlated to IWV. The effect of higher order polynomial terms will be discussed further on. In order to simulate realistic measurements, a random Gaussian noise was added to the modeled brightness temperatures, with a mean and standard deviation of resp. 0 K and 0.5 K ; those values were identified by Küchler et al. (2017) as the characteristics of the measurement noise of the 89-GHz radiometer. Secondly, surface measurements are included (temperature, sea-level pressure and relative humidity); in the case of the radar-radiometer set-up that is used here, a weather station is collocated, meaning those measurements are available at the location of the instruments. The third class of input features comprises geographical descriptors: latitude, longitude, altitude; the day of year is also included in this group of features, as a means to account for seasonal variability in atmospheric and meteorological conditions. When available, a fourth category is added to the input features with reanalysis data (precipitable water and liquid water) from ERA5 (Copernicus Climate Change Service, 2020). The spatial and temporal resolution of this reanalysis data is too low for it to be held as ground truth, but it can serve as a reasonable rough estimate and thus bring some improvements to the statistical learning process – although it could not be included as such in a physical model. Those four groups of features are used both for the retrieval of IWV and that of LWP. In the case of LWP, an additional input feature can be added, which is the output of the IWV retrieval algorithm. The impact of each of those feature groups on the retrieval will be discussed in Sect.5.

### 4.2 Dataset preprocessing

Rain events should be excluded from the training set, since they are out the algorithm's range of validity, as explained in Sect. 3. Profiles with LWP > 1000 $g\,m^{-2}$ are therefore removed (i.e. in the range of heavy rain according to Cadeddu et al., 2017, and in view of the discussion conducted in Sect. 3.2). The resulting dataset contains $\sim 10^6$ profiles and is used for the design of the IWV retrieval algorithm.

### 4.2.1 Further preprocessing for LWP dataset

In the case of LWP retrieval, additional pre-processing is needed, since the forward model produced a large majority of clear-sky cases. If left as such, the training phase would result in a strong bias of the retrieval toward low LWP values (a bias of $\sim 100\,\mathrm{g\,m^{-2}}$ for LWP $> 400\,\mathrm{g\,m^{-2}}$ was noted in the development stages of the algorithm): this is a common artefact in statistical learning algorithms, as an effect of unbalanced training set. In order to avoid this, the dataset was subsampled so that clear-sky and cloudy cases (up to $600\,\mathrm{g\,m^{-2}}$) would be equally represented; the value chosen for this threshold results from a

trade-off between bias reduction and preservation of overall accuracy. The resulting histogram is shown in Fig. 2, and the LWP dataset thus contains $\sim 10^5$ profiles. In the case of IWV, the distribution is also not uniform, but it suffers from a much smaller assymetry than the initial LWP data set. After some trials, it was considered preferable to use the full IWV data set rather than go through subsampling steps, which did not seem to bring significant improvements in this case. It should also be noted here that the additional pre-processing that was necessary for the LWP retrieval algorithm led us to design two separate algorithms,

rather than a single one that would retrieve IWV and LWP at once. Indeed, while LWP retrieval is mostly relevant in cloudy cases, IWV can show some significant variability in clear-sky cases, which should therefore not be excluded from the training stage.

### 4.3 Statistical retrieval using a neural network

After preprocessing, LWP and IWV datasets were randomly split into training, validation and testing set (70 %-15 %-15 %),

and normalized using mean and standard deviation of each input feature in the training set. The validation set is used for tuning the hyperparameters of the neural network, while the final evaluation metrics are computed on the testing set. A densely-connected neural network architecture was chosen over linear regression and decision-tree-based retrieval techniques for it was found to produce more reliable results, with higher accuracy than the former and less prone to overfitting than the latter. The algorithm was designed using the Keras library in Python (Chollet et al., 2015). The neural network was trained through mini-

batch gradient descent, using RMSprop optimizer which allows for learning rate adaptation and is often used for statistical regression problems (Chollet, 2017). As comes across from the training curve of the LWP retrieval on Fig. 3, the training dataset is large enough to ensure that the algorithm is not prone to overfitting: indeed, the error on the validation set quickly drops when the size of the training set, then plateaus with a slight decrease. In other words, the accuracy of the algorithm is not limited by the amount of data used in the training stage. Figure 4 and Table 1 summarize the resulting architecture and

relevant parameters of the algorithm. These include the description of the neural network's structure (number of neurons and hidden layers) as well as training parameters such as the batch size and number of epochs, i.e. the number of iterations through the entire dataset in the learning phase. Different versions of the algorithm were trained, using various sets of input features, to assess the importance of each category (discussed below).

## 5 Results on synthetic dataset

In this section, the algorithm is evaluated on the synthetic dataset (testing set), through different criteria. Overall, results are encouraging and the retrieval appears to be robust. Some limitations can be identified, which will be discussed here. Additionally, an analysis on the impact of the various input features on the retrieval of IWV and LWP is conducted.

### 5.1 Error curves

Fig. 5 presents the distribution of the error on the testing set, for the best version of the algorithm, which is the one that uses the full set of input features. In panels c) and d), the target variable, respectively IWV and LWP, is binned to intervals on which the root mean square error (RMSE) is calculated. This illustrates the behavior of the algorithm across the entire range of values, rather than summarizing the performance with a single metric such as total RMSE, which can conceal specific behaviors related to the distribution of the target variable in the dataset. Along the same line, we emphasize that comparing those total RMSE values to those from other studies should be done carefully because they strongly depend on the dataset from which they are calculated. In a similar way, panel e) (resp. f) illustrates the distribution of the mean bias across the range of IWV (resp. LWP) values. For reference, the definitions of the error metrics that are used in this section and further on are recalled in Appendix A1.

Figure 6 shows how this total error, represented by RMSE (left panels) and by the correlation coefficient (R) (right panels), is affected by the addition or removal of input features. For each set of input features, a full tuning of the algorithm was performed, and the results that are presented correspond to those from the tuned – i.e. best – version on the testing set.

### 5.1.1 IWV algorithm

Overall, the IWV retrieval algorithm yields a RMSE of 1.6 $\mathrm{kg\,m^{-2}}$ on the testing set, which corresponds to a relative error of 6.5 %. For comparison, the ERA5 data alone has a higher RMSE (3.4 $\mathrm{kg\,m^{-2}}$) on the same data set. Looking at Fig. 5 a), c) and e), it comes across that the retrieval performs quite well over the full range of IWV values, and the error distribution is relatively homogenenous. For high IWV values, however, a significant negative bias is present (as large as -6 $\mathrm{kg\,m^{-2}}$). Because such high values are underrepresented in the dataset, they are not well captured during the statistical learning stage, which leads to a systematic underestimation. However, these are by definition "border" cases, for which a decrease in accuracy is to be expected.

From Fig. 6 a) and b) it comes across that the IWV retrieval is significantly improved by the addition of multiple input features. The highest accuracy is obtained with the full set of input t features, and corresponds to a RMSE of 1.53 $\mathrm{kg\,m^{-2}}$. On the other hand, including solely $T_B$ measurements in the input deteriorates the RMSE to nearly 6 $\mathrm{kg\,m^{-2}}$. If only one input feature were available, all the versions would predict worse results than those given by reanalysis data. Including $T_B$ in the retrieval does not lead to the same leap in accuracy than for LWP (discussed in the following subsection); however, excluding $T_B$ from the input features degrades the RMSE to 2.56 $\mathrm{kg\,m^{-2}}$, i.e. + 67 % error, which clearly shows that brightness temperature incorporates

additional relevant information into the retrieval.

An analysis was conducted to identify the importance of higher order polynomials in the algorithm, a summary of which can be found in Appendix A1. It was found that the most accurate retrieval is obtained by including $T_B$ and $T_B^2$. If higher order terms are added, this slightly reduces the accuracy of the retrieval, and also degrades its robustness to $T_B$ miscalibration. On the other hand, including only $T_B$, while it makes the algorithm slightly more stable, does not appear as the best solution for it has lower accuracy. Hence, the results which are presented here and in the following sections are those obtained using $T_B$ and $T_B^2$.

### 5.1.2 LWP algorithm

The LWP retrieval algorithm has a RMSE of 86 $\mathrm{g\,m^{-2}}$ at best on the testing set (training set: 84 $\mathrm{g\,m^{-2}}$ and validation set: 86 $\mathrm{g\,m^{-2}}$). This corresponds to a relative error of 29 % on the testing set. Let us underline that the subsampling which is performed on the dataset for the retrieval of LWP is applied to training, validation and testing sets: the results that are presented here are therefore computed on the testing set with a truncated distribution – i.e. after subsampling. Additionally, if clear-sky cases are removed using 30 $\mathrm{g\,m^{-2}}$ as a threshold value, following Löhnert and Crewell (2003), the relative error is 18 %. As already mentioned, the total RMSE values given here should be taken with care since they depend on the data set's distribution. For comparison, when the retrieval is implemented on the full dataset, i.e. without the subsampling step, the total RMSE drops to 40 $\mathrm{g\,m^{-2}}$. The RMSE is here again rather homogeneous across the range of LWP values (Fig. 5.d), with however a small bias of around 20 $\mathrm{g\,m^{-2}}$ for low LWP values (visible in Fig. 5 f), which are slightly overestimated, and while there is an underestimation of large LWP (LWP > 800 $\mathrm{g\,m^{-2}}$), with a negative bias down to -100 $\mathrm{g\,m^{-2}}$. Both biases result from an effect of regression towards the mean, which is a intrinsic artefact of statistical algorithms. The significant negative bias for large LWP values is enhanced by the lack of data in this range. It is likely acceptable, for it would correspond mostly to raining cases (light to moderate) which the retrieval does not aim to capture; yet this highlights once again that those cases are out of the algorithm's scope and that retrievals with high LWP should be taken with care.

The analysis of higher order terms' importance in the case of LWP retrieval shows that the best results are obtained by using $T_B$ polynomials up to the fourth order (see Appendix A2), while this does not affect significantly the stability of the retrieval to errors in $T_B$. Let us highlight that in the case of a linear regression, one would expect the error to diverge when high-order polynomials are included. This is not the case here, because of the saturating behavior of the neural network. Therefore, in the results which are shown here and further, "$T_B$" implies that $T_B$ polynomials up to the fourth order are used.

Figures 6 c) and d) show that for LWP retrieval, input features other than $T_B$ only bring second-order improvements, while they were shown to be crucial in the IWV retrieval. For instance, the addition of reanalysis data significantly improves the IWV retrieval, but only in a relatively minor way does it increase the accuracy of LWP retrieval. On the contrary, excluding $T_B$ from the input features leads to RMSE near 200 $\mathrm{g\,m^{-2}}$ and R< 0.7, i.e. to values with make the retrieval not relevant. This highlights that while environmental descriptors are well correlated to IWV, they are not sufficient to provide a reasonable estimate of LWP, for which microwave radiometer measurements are critical. An additional reason for this high dependence

on $T_B$ is that LWP at a given location can have a large temporal variability due to cloud dynamics in the atmospheric column, which might not always be captured in the time series of surface atmospheric variables, nor by ERA5 models which have a comparatively low spatial and temporal resolution.

Still, the accuracy of the algorithm drops severely when no other features are considered than brightness temperature (RMSE of 140 g m$^{-2}$). This means that, albeit second-order when taken individually, and somehow redundant when all used together, the
secondary input features are efficient in incorporating statistical trends and climatological information to the retrieval during the training phase.

Adding IWV prediction as an input feature to the LWP retrieval has a very minor impact. For clarity, it was only included in Fig. 6 c) in the best-case scenario and not for every other combination of input features. This is not surprising, since it is itself the output of an algorithm that relies on essentially the same input features. However, the slight improvement that is seen can
be understood by recalling that the IWV retrieval algorithm was trained on a much larger dataset, which includes in particular a larger number of clear-sky cases (cf. Sect. 3).

## 5.2 Sensitivity to instrument calibration

In order to assess the stability of the algorithm with respect to potential miscalibration or calibration drift of the radiometer, $T_B$ offsets were virtually added to the testing dataset before implementing the retrieval. Figure 7 illustrates the behavior of
the algorithm when such a miscalibration, with a constant offset is present (varying from 0 to 5K). Panel a) shows that a 5 K offset in $T_B$ results in a 30 % increase in RMSE for the IWV estimations, which is non-negligible. Ensuring proper radiometer calibration thus seems crucial in constraining the error of this retrieval. For comparison, the 89-GHz radiometer presented in Küchler et al. (2017) has a nominal accuracy of 0.5 K, after calibration. If the calibration cannot be ensured, and if there is no means to correct for miscalibration (of > 3K), it is preferable for IWV retrieval to use the algorithm that does not rely on $T_B$,
shown with the black dashed line.

In terms of relative impact, the LWP algorithm is less affected (Fig. 7 b) ) with an increase of the RMSE of less than 10 % for an offset of 5 K in $T_B$, which makes it reasonably stable to inaccuracy of $T_B$ measurements. It also appears that the different versions are affected in a similar way by offset $T_B$ values. However, the algorithm which includes the prediction of
IWV in the input features diverges faster than the others. This is understandable, for the error on $T_B$ propagates through the IWVpred input feature, in addition to the $T_B$ features themselves. Therefore, in the case of uncertain calibration, more robust results would be obtained without including this feature.

It is noteworthy that for $T_B$-only retrievals, the addition of a $T_B$ offset does not result in a large increase of the error: for
IWV, the addition of a 5K offset increases the RMSE from 5.6 to 6.2 kg m$^{-2}$; for LWP, the same offset leads to an increase from 139 to 142 g m$^{-2}$. This behavior is also observed when looking at how the bias, instead of the RMSE, increases with the addition of a $T_B$ offset (not shown). In both cases, the error increases more drastically when multiple features are included, than when only $T_B$ is used as input. One possible explanation for this effect is the following: when incorporating numerous

input features, the algorithm is able to narrow down the range of possible IWV and LWP values in a given environmental context; in this constrained configuration, the correlation and sensitivity of the retrieval to $T_B$ are then enhanced, leading to a stronger influence of a $T_B$ offset.

## 5.3 Geographical distribution of the error

One of the motivations of this study was to design an algorithm that could be used across the globe with a constrained uncertainty. Figure 8 illustrates the geographical distribution of the error for both LWP and IWV retrievals, using the synthetic radiosonde-based dataset. Two approaches were used to assess this error: first, RMSE values were calculated on the entire set of data available for each location, excluding LWP greater than $1000 \text{ g m}^{-2}$. Second, the RMSE was normalized by the mean value of LWP (resp. IWV) for each site, excluding low values (LWP less than $20 \text{ g m}^{-2}$, i.e. using a conservative threshold to exclude clear-sky cases). Note that this normalized error is not equal to the relative error; rather, it gives an idea of how large the RMSE of the retrieval is, compared to the mean values that are observed at a given location.

From the non-normalized error (left panels of Fig. 8), it can be seen that most high- and mid-latitude locations have a constrained RMSE around 20-60 $\text{g m}^{-2}$, while tropical sites are not as well captured, with RMSE exceeding 120 $\text{g m}^{-2}$ in some locations. The temperature and humidity conditions, as well as the strong precipitation events that typically occur in those regions, are probably responsible for this discrepancy. Cases with high LWP are more common under such climatic conditions, and it was observed in Sect. 5.1.2 that the accuracy of the algorithm decreases in that range. Tropical climates are underrepresented in the dataset, for little data is available from this region in comparison with mid-latitude areas: their specificity might therefore not be fully captured during the learning stage of the algorithm. This accounts at least partly for the enhanced error over the Indian peninsula and South-Eastern Asian islands.

The normalized error (right panels of Fig. 8) shows that the error is overall of the same order of magnitude across the globe. However, a few regions stand out from this analysis, which typically feature arid climates: the stations of Dalanzadgad (Mongolia), Salalah (Oman), Minfeng (China, north of Tibet), Jeddah (Saudi Arabia) all have a normalized error on LWP higher than 0.7, and are in the desert. In a similar way, it appears that the IWV retrieval algorithm performs poorly – in terms of normalized error – in cold environments where absolute humidity is low, as in Sermersooq (Greenland). In such regions, the new algorithm is not sensitive enough to capture accurately the fine variations of atmospheric vapor and liquid water content: if detailed studies of those areas were to be conducted, more than one radiometer frequency would likely be necessary, along with specific training sets on which to perform the statistical learning, as was done in the Arctic by Cadeddu et al. (2009).

## 6 Evaluation in two contrasted datasets

As a further step in the validation process, the algorithm was applied to data from two campaigns involving WProf, first in Payerne, Switzerland, then near Pyeong-Chang, South Korea (see Sect. 2 for the full description of the datasets). In both cases,

the output of the retrieval is compared with values retrieved through other methods, either a multi-channel radiometer or – in the case of IWV – radiosonde data.

## 6.1 Payerne 2017

### 6.1.1 IWV retrieval

The results of the new IWV retrieval algorithm are compared to those from MeteoSwiss' operational radiometer HATPRO, and to the radiosonde-derived values. From Fig. 9 a) and c) it appears that the IWV retrieval has relatively limited spread but has a constant bias (-1.8 $\mathrm{kg\,m^{-2}}$), which is visible both in the comparison against HATPRO (a) and radiosonde-derived measurements (c). This might be due to a bias in ERA5 data during this timeframe over the region (with a value of -4.1 $\mathrm{kg\,m^{-2}}$), which is visible in ERA5 records during the entire campaign (not shown here) and for which there is no clear

explanation at this stage. This bias points to one of the limitations of the IWV retrieval algorithm, which is sensitive not only to radiometer miscalibration but also to possible biases in other input variables, which can be difficult to monitor and assess – as in the case of ERA5 values in Payerne. In spite of this, the top panels of Fig. 10 (which illustrate the error vs. HATPRO measurements) and Fig. 11 (error vs. value derived from radiosounding) show that overall, the implementation of the different versions of the algorithm on the Payerne dataset matches the conclusions from the testing set results: more features lead to an

enhanced precision of the retrieval. The accuracy drops when only one or two groups of input features are included, but no single group of features seem to increase the accuracy alone. There is however a difference between Fig. 10 b) and Fig. 11 b): in the latter, higher R (and similar RMSE) is actually obtained from the algorithm that does not use $\mathrm{T_B}$ in input, than with the full set of input features. This is at first surprising, but it was explained by taking a closer look at the results: the algorithm without $\mathrm{T_B}$ leads to IWV values which are more smooth and less sensitive to short-time variations. These are not reflected in

the comparison against radiosonde data, for which a 30-min averaging was implemented.

When variations over a small timeframe are considered, the inclusion of $\mathrm{T_B}$ improves the retrieval, as comes across from the comparison against HATPRO's measurements in Fig. 10.

### 6.1.2 LWP retrieval

Figure 9 b) shows that LWP values retrieved with the new algorithm are in general agreement with those obtained thanks to

400 HATPRO, although a larger spread is observed than in the IWV retrieval. A saturation effect can be seen near precipitation onset, when LWP values from HATPRO reach 600 $\mathrm{g\,m^{-2}}$. Additionally, outliers are visible as vertical and horizontal bars close to the axes, for which two hypotheses are considered. One is that the distance between the two instruments was big enough, that in some cases a liquid water cloud would overpass one of the two instruments but not the other. Hence, HATPRO would measure a non-zero LWP while WProf would indicate a clear sky, or vice-versa. Besides, measurement artefacts cannot

be excluded, e.g. due the persistence of a liquid water film on the radome of either radiometer, after precipitation or due to condensation.

For comparison, the method described in Küchler et al. (2017) was implemented (further on referred to as K17), by performing a quadratic regression on a dataset consisting solely of radiosonde profiles collected in Payerne. As proposed by the authors, a first version (K17A) relies on a measurement vector consisting of $T_B$, $T_B^2$, as well as the IWV estimate from reanalysis data $IWV_{ERA5}$ and $IWV_{ERA5}^2$. Another version (K17B) includes only $T_B$ and $T_B^2$. Theoretical RMSEs derived for those quadratic regressions on the synthetic dataset (19 720 profiles) are $21 \, \mathrm{g \, m^{-2}}$ and $43 \, \mathrm{g \, m^{-2}}$, respectively, which is similar to the values obtained by the authors on radiosonde data from De Bilt (the Netherlands), i.e. $15 \, \mathrm{g \, m^{-2}}$ and $44 \, \mathrm{g \, m^{-2}}$.

K17A and K17B were applied to Payerne campaign dataset, and their results are compared to those from the new algorithm in Fig. 10. The error metrics are calculated using HATPRO's values as a reference. The algorithms perform in a similar way, with slightly better results for the new algorithm when at least one of the secondary input features is included. We remind that K17A and K17B were specifically tuned on Payerne data, while the new algorithm was tuned globally, on a dataset that did not comprise radiosonde profiles from Payerne.

## 6.2 ICE-POP 2018

As detailed in Sect. 2, the South Korean deployment of WProf in 2017-2018 also offers an opportunity to compare results from the IWV retrieval to IWV from radiosonde measurements.

The analysis of the $T_B$ timeseries showed that a miscalibration of the radiometer led to unrealistic – negative – values for which a correction had to be implemented, through the addition of a constant offset to $T_B$ measurements. The value of this offset (20 K) was determined by computing theoretical brightness temperatures from clear-sky radiosonde profiles and comparing them to measured $T_B$s, following the approach of Ebell et al. (2017). This is however only a first-order correction whose output should be taken with care, especially after the analysis of Sect. 5.2 which underlined the importance of $T_B$ accuracy for IWV retrieval.

After this correction, the IWV retrieval gives coherent results (see Fig. 12), with a total RMSE that is slightly lower than that obtained on the testing data set ($1.25 \, \mathrm{kg \, m^{-2}}$). The best results are found when several input features are included and drop severely when no secondary input features are used, which corresponds to the results on the synthetic data set presented in Sect. 5. The algorithm largely relies on non-radiometric features, and this is even more the case in cold and dry environments like that of ICE-POP, where IWV is low. In fact, slightly better results are obtained with all input features except brightness temperature. The miscalibration of the radiometer, which may not have been perfectly corrected by the addition of a constant offset, might emphasize this error. This also corresponds to what was noted in Payerne: when the results are averaged over 30 minutes, brightness temperature brings little, if any, improvement to the results. $T_B$ is relevant when a higher temporal resolution is considered (cf. Sect. 6.1.1) – for which no comparison was available during ICE-POP – or when ERA5 data is significantly off. In this case however, it comes across from Fig. 12 that the algorithm is consistently outperformed by ERA5 products: they have both a lower RMSE and a higher R, which makes the algorithm less relevant for the study of this specific campaign. The high accuracy of ERA5 data during ICE-POP also explains the high correlation coefficient of the retrieval that uses ERA5 and Geographical input features: since the geographical parameters are constant, the temporal variability is that of the reanalysis data and therefore the correlation coefficient of the retrieval is close to that of ERA5 data alone. Let us highlight

that although reanalysis data outperforms the retrieval for ICE-POP, this was not the case in Payerne nor in the full radiosonde data set, where the algorithm has a higher accuracy than ERA5 values. Possibly, the dry and cold weather that was observed during the ICE-POP campaign featured little short-term variability and was associated with stable atmospheric conditions that were particularly well captured in ERA5 reanalyses. Snowfall events during the campaign, as well as occasional fog, can also

bias the retrieval by enhancing brightness temperature.

The analysis of the ICE-POP data was taken a step further to explore the latter point. It appears that the IWV retrieval is most reliable in non-precipitating or cold conditions, i.e. when little liquid water is expected in the column. To visualize this, periods with no precipitation or fog are identified using WProf's radar measurements as time steps with low radar equivalent

reflectivity (Ze < -10 dBZ) in the lower gates (first kilometer above the radar), and temperature time series are provided by the weather station coupled to WProf. Fig. 13 shows the scatter plot of the error – for the algorithm that includes all input features – color-coded in a way to differentiate dry from precipitating or fog conditions: black triangles correspond to dry timesteps, and circles to timesteps with Ze > -10 dBZ, with their color indicating surface temperature. The algorithm yields a larger bias in rain – as was expected in the design steps of the algorithm (Sect. 3) – but also during snow events with relatively warm

temperatures, close to or slightly above 0 °C (Fig. 13). Changes in the dielectric properties of snowflakes during the melting process can explain this increased error; additionally, the process described by Kneifel et al. (2010) and which was recalled in Sect. 3 suggests that snowfall events with large snow particles (typically present with relatively mild temperatures) could have a non-negligible contribution to brightness temperature, which might explain the enhanced error in those cases.

## 7    Summary and conclusions

A new site-independent method was designed for the retrieval of LWP and IWV from a single-channel ground-based radiometer. In addition to 89-GHz brightness temperature, additional input features were used for the retrieval, such as surface atmospheric variables (temperature, pressure and humidity) and information on the geographical location and season. A neural network architecture was chosen for the statistical learning.

Training and testing were performed on a synthetic data set that was built using radiosonde profiles worldwide. The geograph-

ical distribution of the error shows that the algorithm performs better in mid-latitudes and regions with a moderate climate than in areas with a extreme climates – either arid or very moist – which include both tropical and polar regions, that are not well represented in the training dataset due to lack of available data. Also, the forward model that was used should most likely be revised in order to capture finely the atmospheric conditions in such specific environments. In addition, the training dataset lacks data from locations with complex orography, and more in-depth investigations should be conducted regarding the

reliability of the retrieval in such terrain (Massaro et al., 2015).

The algorithm was then applied to two contrasted data sets, one reflecting summertime weather conditions in Switzerland, and the other in winter conditions in South Korea. For this application, measurements from RPG's cloud radar-radiometer system were used.

In Payerne, the new LWP retrieval was found to perform slightly better than the method proposed by Küchler et al. (2017) for the same instrument, albeit the latter algorithm was specifically trained using radiosonde data from Payerne. When compared to radiosonde measurements of IWV, the IWV retrieval was found to be less accurate than that of a state-of-the-art multi-channel radiometer (HATPRO), although both instruments yield errors within the same order of magnitude. In the South-Korean winter dataset, the IWV retrieval proved relatively robust, in spite of a slight bias during some snowfall events, that could be related to the scattering properties of snow particles, which were not taken into account in the forward model. In the case of ICE-POP, reanalysis data was actually more accurate than the IWV retrieval when compared with radiosonde measurements, but its temporal resolution remains low, which makes the use of the algorithm still relevant for retrievals where a high temporal resolution is required.

Further steps in the improvement of the current algorithm would include coupling information from the radar and the radiometer channel (Ebell et al., 2010; Cadeddu et al., 2020). The detection of clear-sky cases with radar data (Mätzler and Morland, 2009) could help monitor the calibration of the radiometer, and introduce $T_B$ offsets for correction when necessary (Ebell et al., 2017). If available through a separate sensor such as a GPS receiver, independent IWV measurements could be included in the algorithm, possibly leading to an enhanced precision of the LWP retrieval. Radar moments could be used to distinguish cloudy from drizzling or rainy cases, similar to the approach conducted by Cadeddu et al. (2020), and to use appropriate DSDs for each case to account for non-Rayleigh scattering by precipitating droplets. However, forward modeling of radar data requires further assumptions on microphysical properties and atmospheric conditions, for which generalization to a global geographical scale is a real challenge. Additionally, the retrieval that is presented here uses reanalysis data as an optional feature, which was shown to be valuable. In the case where near real-time retrievals were necessary, the user could choose to use a version of the algorithm that does not rely on ERA5, but this would be detrimental, especially for the IWV retrieval. Another option would be to implement the algorithm with the output of forecast models (IWV and LWP) instead of reanalysis data. This approach was however not tested as this stage.

Overall, the LWP and IWV retrieval methods that were designed within this study were shown to be robust, both when applied to synthetic and to real datasets, although their performance is inevitably lower than that of multi-channel radiometers specifically designed for LWP and IWV retrieval. While retrieving IWV based on $T_B$ at 89 GHz alone does not lead to accurate results – this would require the use of other microwave frequencies, more suited to the emission spectrum of water vapor – this study showed that reliable retrievals could be achieved by including surface and geographical information, as well as reanalysis data if available, among the input features. The new algorithms should be seen as a valuable tool for atmospheric liquid water and vapor monitoring in the context of radar-radiometer studies. They are non-site specific, and thus do not require further tuning before use on a new site, which makes them easy to implement, while their accuracy is well characterized.

*Code availability.* The code developed in this study can be made publicly available upon request to the authors.

*Author contributions.* ACB and AB designed and conducted the study. ACB processed the data and run the analyses. ACB prepared the manuscript with contributions and supervision from AB.

*Competing interests.* The authors declare that they have no competing interests.

*Acknowledgements.* This project has received funding from the European Union's Horizon 2020 research and innovation programme under grant agreement No 824310. Additionally, the authors are greatly appreciative to the participants of the World Weather Research Programme Research and Development and Forecast Demonstration Project International Collaborative Experiments for Pyeongchang 2018 Olympic and Paralympic Winter Games (ICE-POP 2018), hosted by the Korea Meteorological Administration. In particular, we are thankful to the High Impact Weather Research Center of the Korea Meteorological Administration for providing us with radiosonde data from Daegwallyeong, South Korea. We are grateful to Kwonil Kim and GyuWon Lee from Kyungpook National University for their help in the installation of the W-band cloud radar on site. We would further like to thank Maxime Hervo and Giovanni Martucci from MeteoSwiss for providing us with radiometer and radiosonde data from Payerne, Switzerland. Special thanks go to Josué Gehring, Alfonso Ferrone and Christophe Praz for the deployment of the instruments on the field. Finally, we are thankful to the three anonymous reviewers whose thorough comments helped improve this study.

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

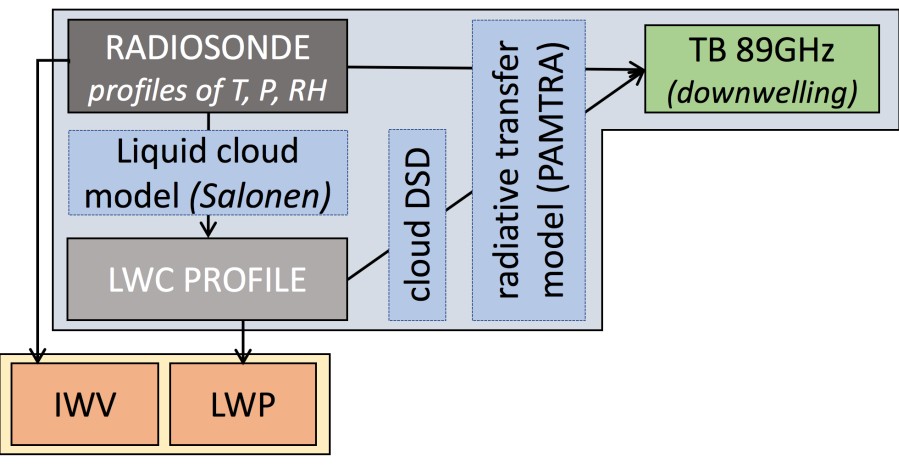

**Figure 1.** Illustration of the different steps of the forward model.

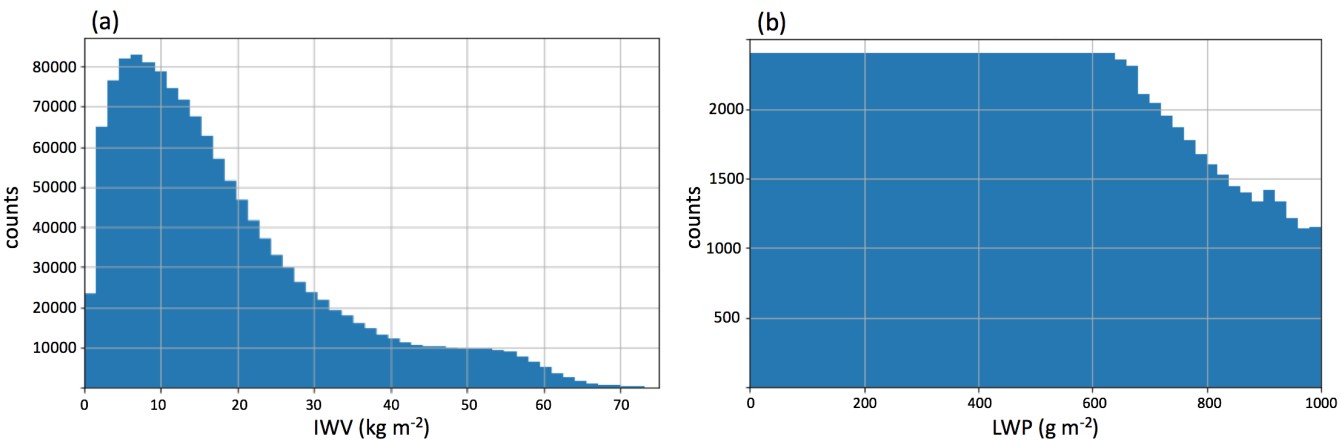

**Figure 2.** Distribution of the target variables (IWV and LWP, resp. in panels (a) and (b)) in the synthetic dataset, after preprocessing.

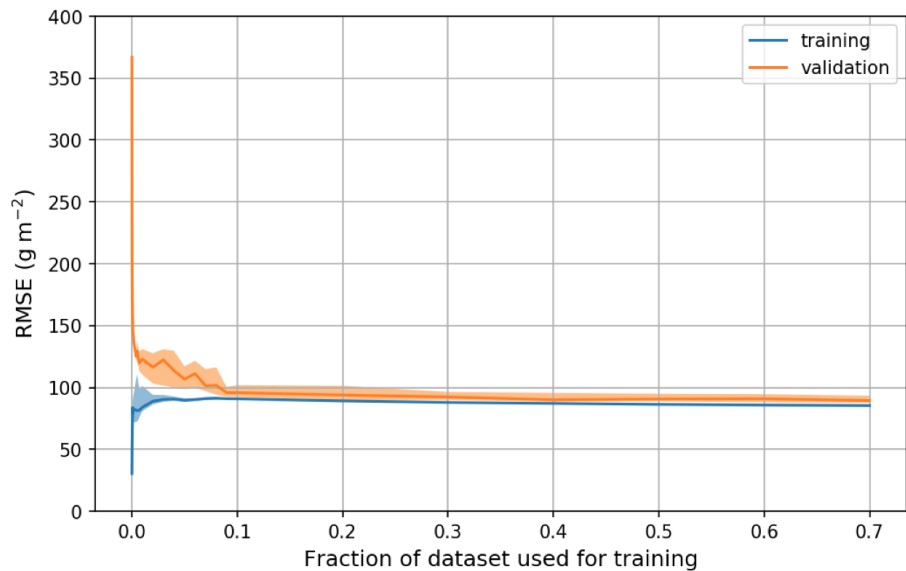

**Figure 3.** Learning curves for the LWP retrieval, showing the RMSE on training and validation set with varying training set size. Shaded areas correspond to the interquartile range calculated over 50 realizations of random splitting of the dataset into training and validation sets, bold lines are the median.

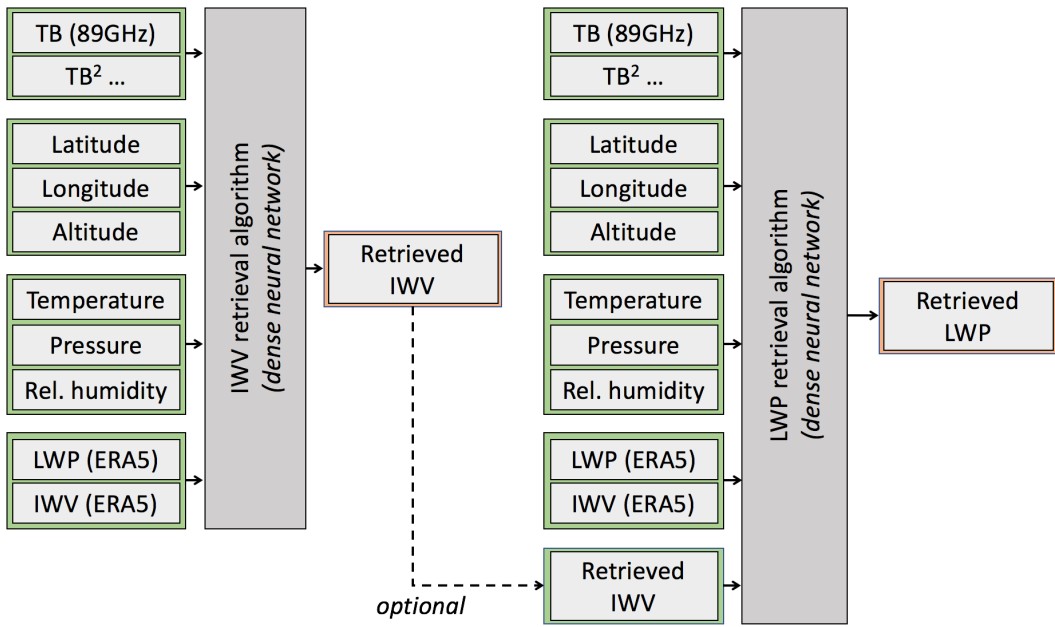

**Figure 4.** Structure of the retrieval algorithms. Some versions of the LWP retrieval include, among the input features, the output of the IWV retrieval. Note that the IWV and LWP algorithms are trained on different datasets.

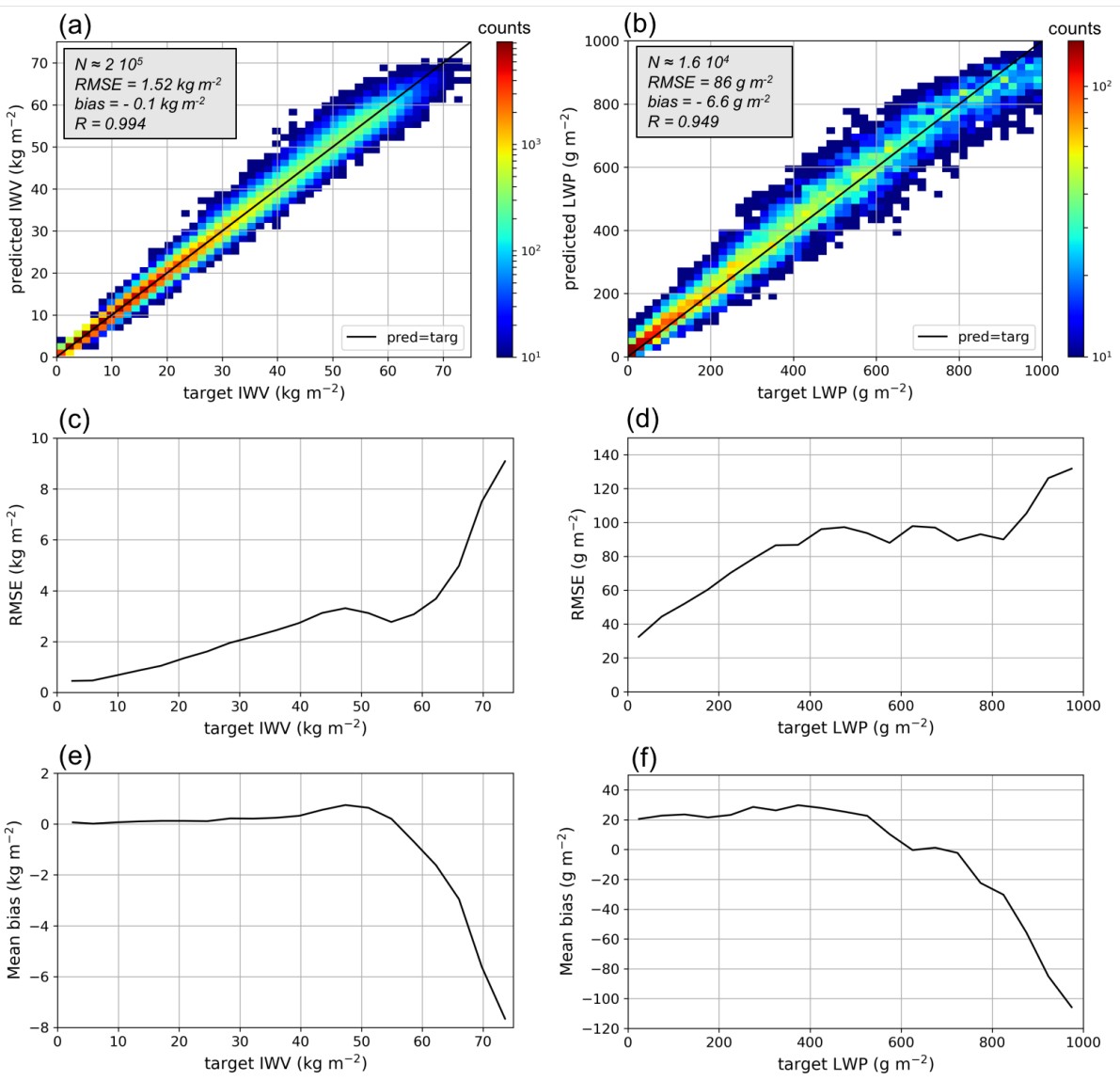

**Figure 5.** Results of the retrieval algorithms on the synthetic testing dataset. The best versions of the algorithms are presented, i.e. the ones which use the full set of input features. Panels (a) and (b) show the distribution of predicted vs. target values of resp. IWV and LWP. The size of the testing set is indicated (N) as well as relevant error metrics (RMSE, bias, R). Panels (c) and (d) illustrate the distribution of the RMSE across the range of IWV and LWP values, binned into intervals of resp. 4 kg m$^{-2}$ and 50 g m$^{-2}$. Similarly, panels (e) and (f) show the distribution of mean bias across the range of IWV and LWP values.

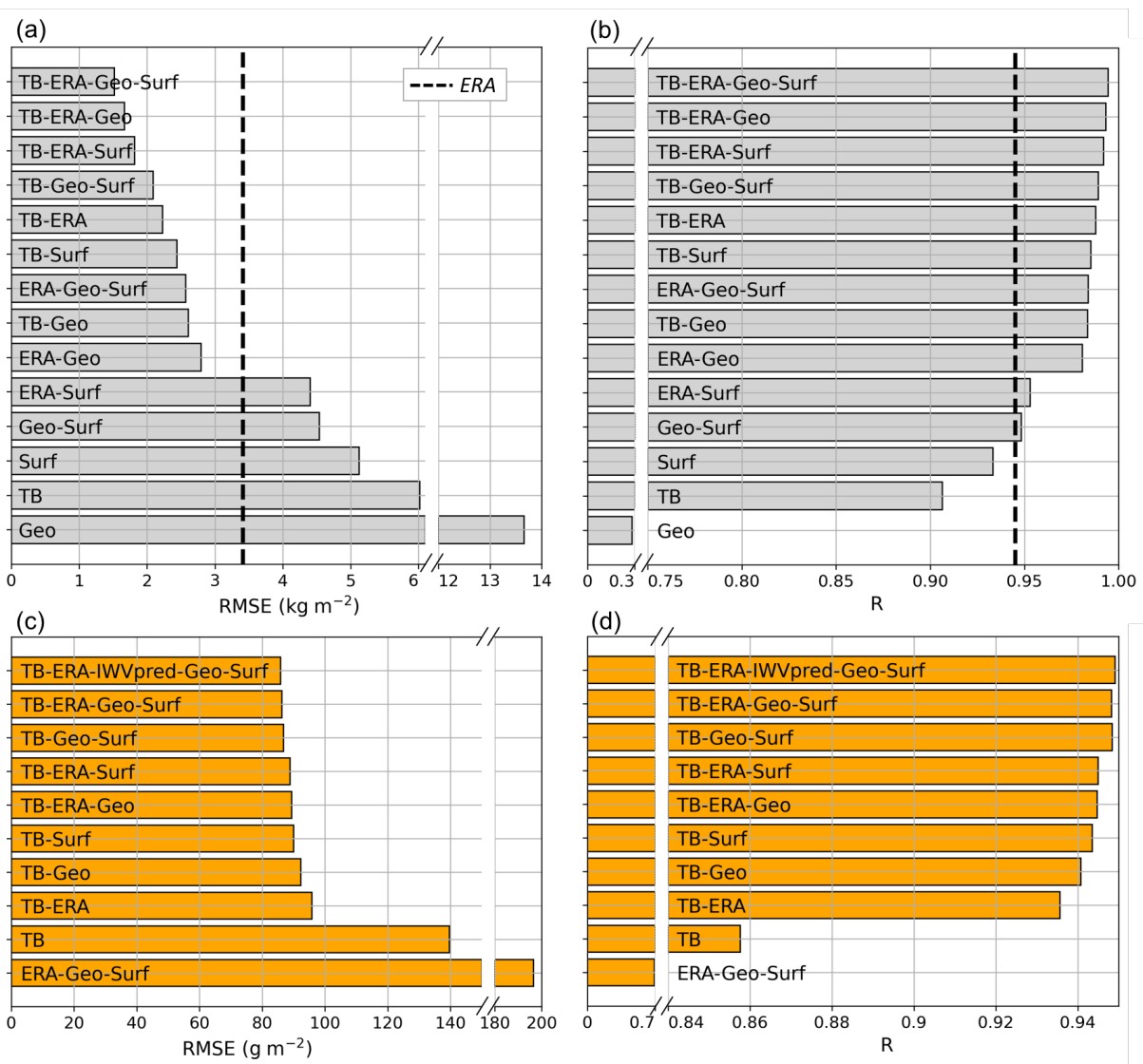

**Figure 6.** Global error metrics (RMSE on the left panels and correlation coefficient R on the right) computed on the testing set for different versions of the (a,b) IWV and (c,d) LWP retrievals. Each bar shows the result of a version whose input features are specified in the label. For example, "ERA-IWVpred-Geo-Surf" corresponds to the version of the LWP retrieval algorithm that uses the following categories of input features: ERA5 variables, IWV obtained from the IWV retrieval, geographical information and surface measurements. The bars are sorted with increasing RMSE. For the IWV retrieval, the accuracy of the algorithm is compared to that of reanalysis data alone (dashed lines).

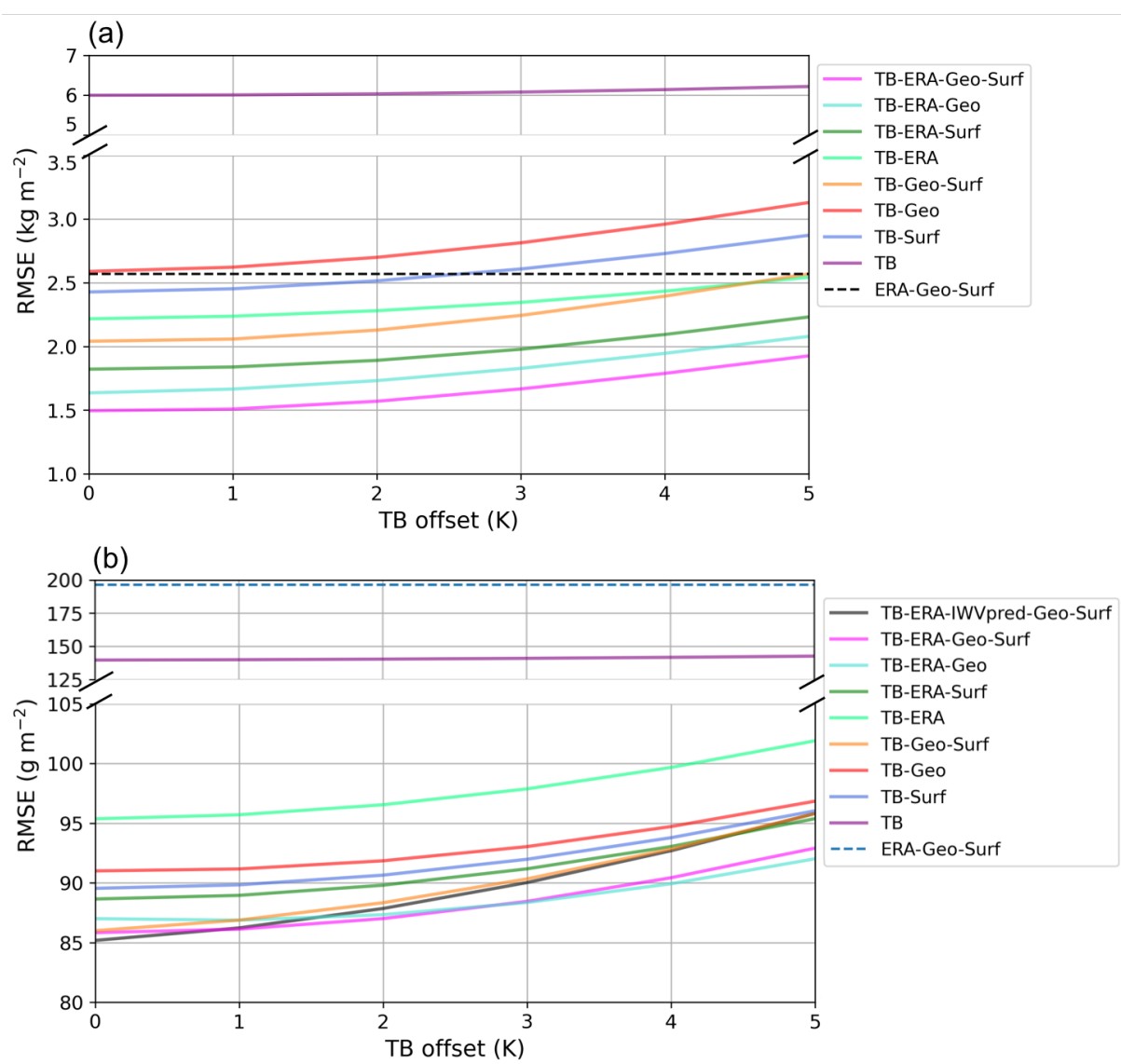

**Figure 7.** RMSE on testing set of the different versions of the (a) IWV and (b) LWP retrieval, after addition of a constant $T_B$ offset in the input. Dashed lines show the retrievals without $T_B$ in the input features.

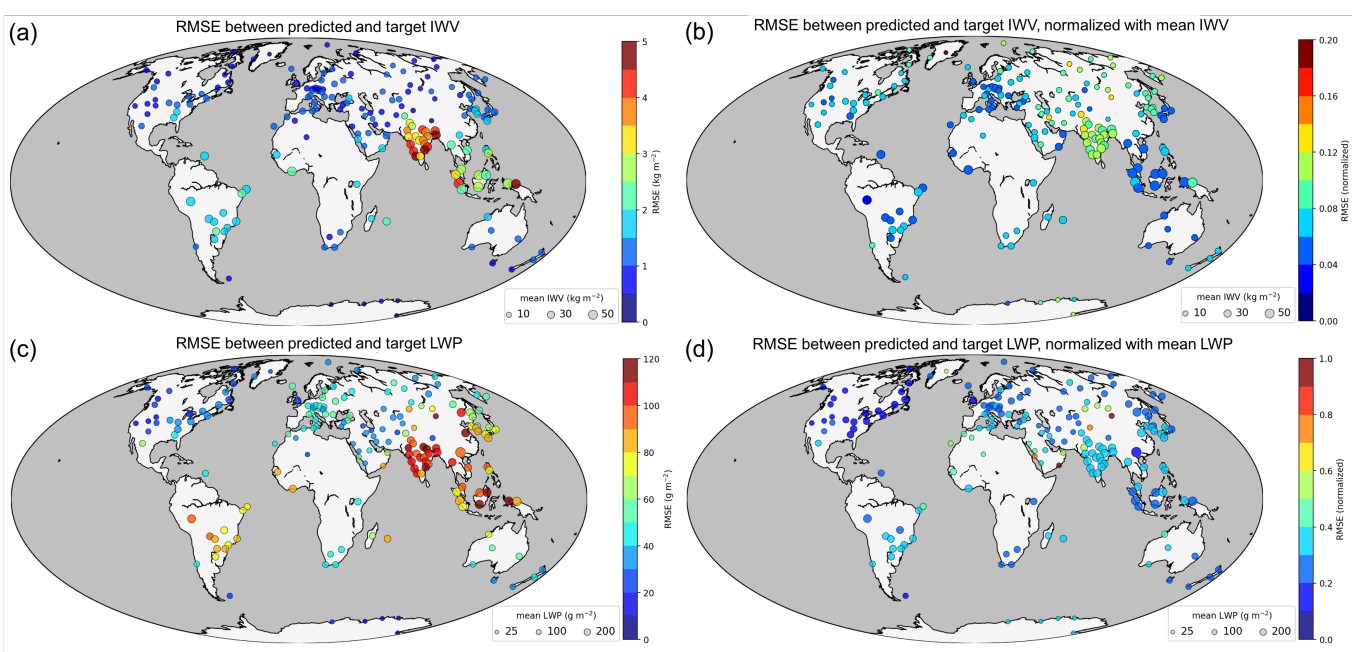

**Figure 8.** Geographical distribution of the error on the synthetic dataset. Left panels (a) and (c) illustrate the total RMSE on resp. IWV and LWP. In panels (b) and (d) is shown the normalized error, i.e. the RMSE normalized by the mean value of IWV (resp. LWP) at each location. For the evaluation of LWP, clear-sky as well as strong rainy cases are removed (resp. LWP $< 20 \, \mathrm{g \, m^{-2}}$ and LWP $> 1000 \, \mathrm{g \, m^{-2}}$). The size of the disks represents the mean value of IWV or LWP at each site, while the color codes for the error of the retrieval.

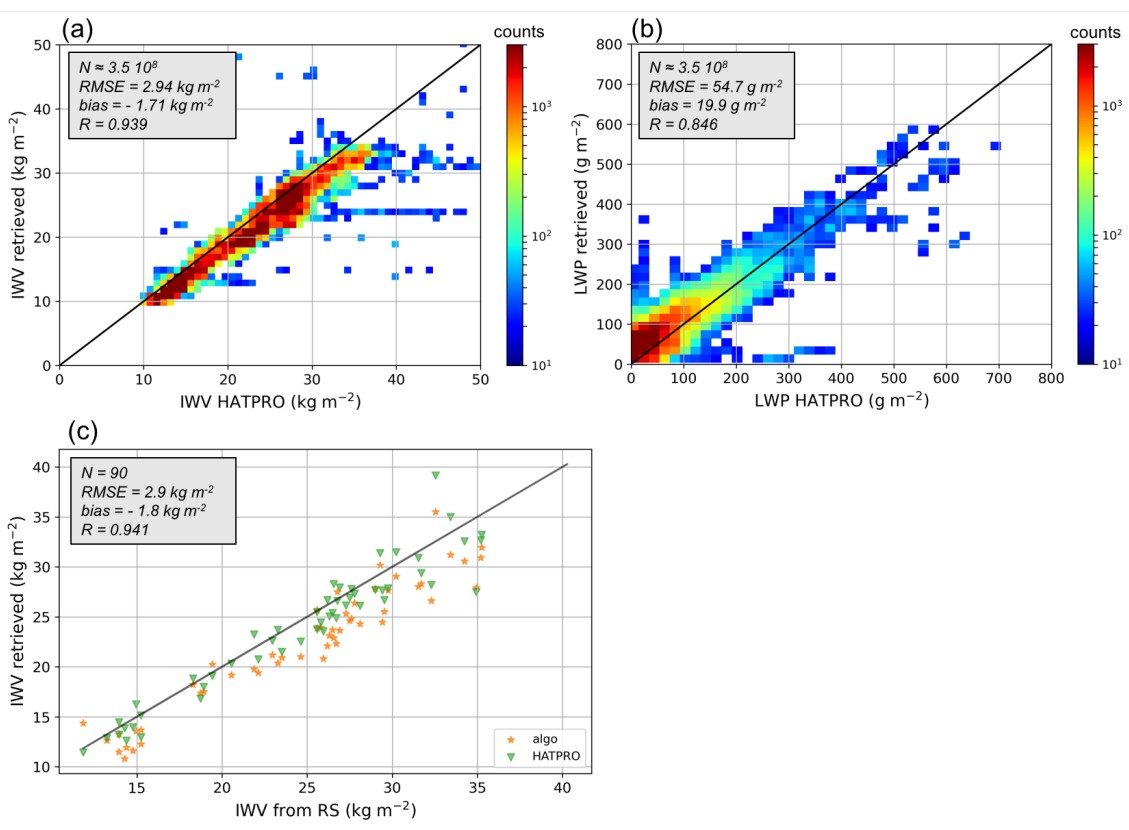

**Figure 9.** Comparison of (a) IWV and (b) LWP retrieved over Payerne with the new algorithm, using the full set of input features, against the retrieval from MeteoSwiss' radiometer HATPRO. Panel (c) shows IWV retrieved from the new algorithm and from HATPRO against the from radiosonde measurements; a 30 minute time averaging is used for radiometer measurements. The size of the data set is indicated (N) as well as relevant error metrics (RMSE, mean bias, R).

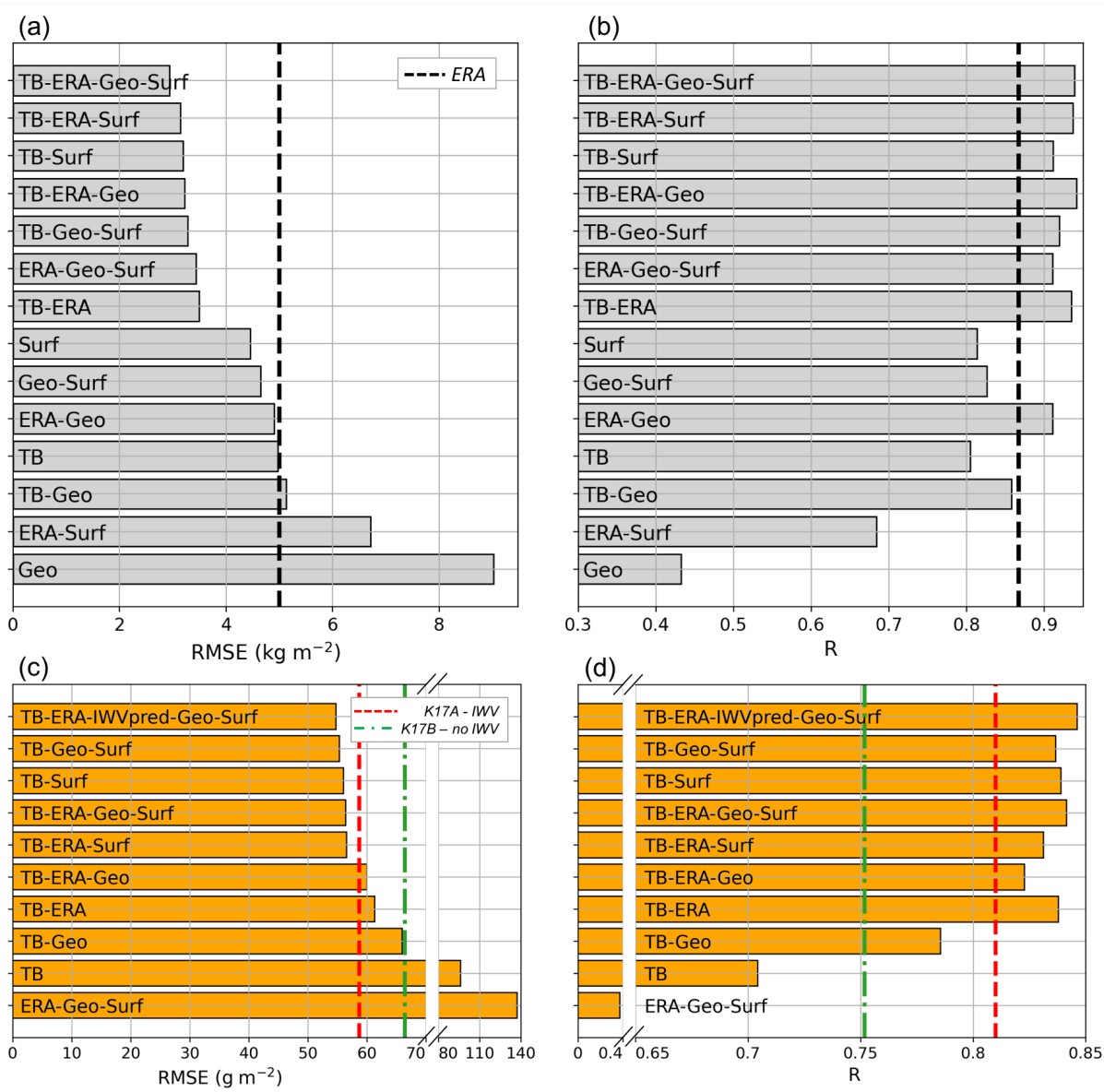

**Figure 10.** Error of the new retrieval algorithms over Payerne compared to HATPRO retrievals. In panels (a, c) the RMSE of resp. IWV and LWP is calculated for different versions of the algorithm. Similarly, R is shown in panels (b, d). Each bar shows the result of a version whose input features are specified in the label. In panels (a, b), the black dashed line shows the error of IWV from ERA5 reanalysis data. In (c, d), the dashed lines present the results from K17A and K17B, as defined in the body text.

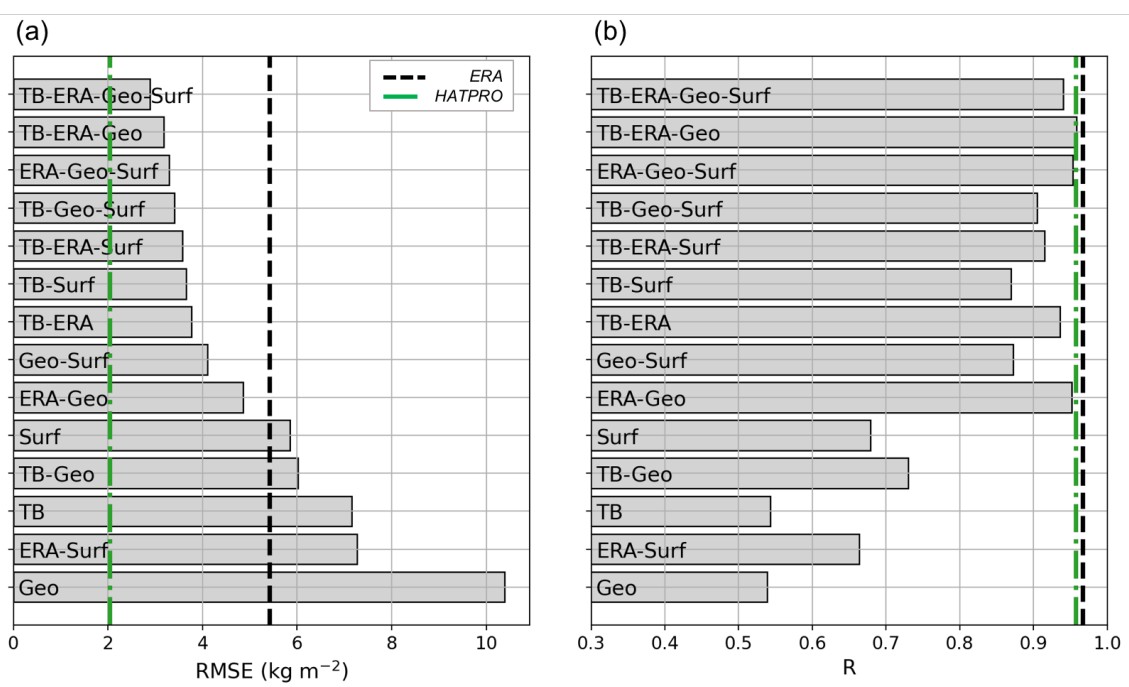

**Figure 11.** Results of the IWV retrieval in Payerne compared to radiosonde measurements (a: RMSE and b: R). The radiometer measurements are averaged over 30 minutes. For comparison, the dashed lines illustrate the error of HATPRO (green) and ERA5 (black) vs. radiosonde-derived IWV.

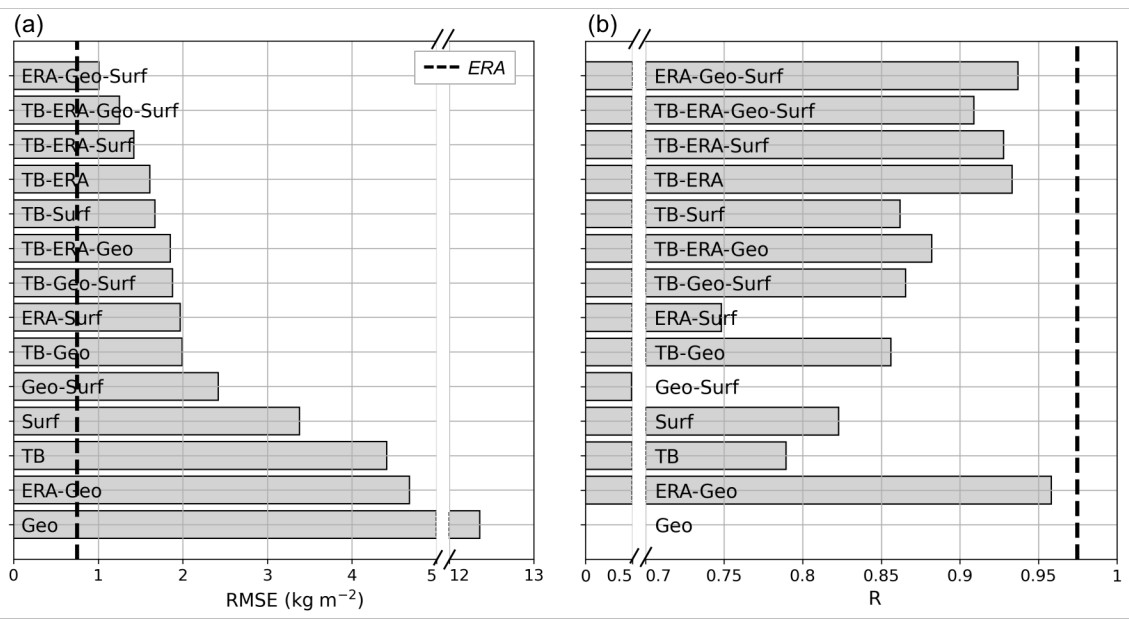

**Figure 12.** Error of the IWV retrieval during the ICE-POP campaign with different versions of the algorithm. The RMSE is computed against IWV from radiosonde profiles, after 30 minutes of temporal averaging in the radiometer data. The dashed line shows the error of IWV from ERA5 reanalysis data.

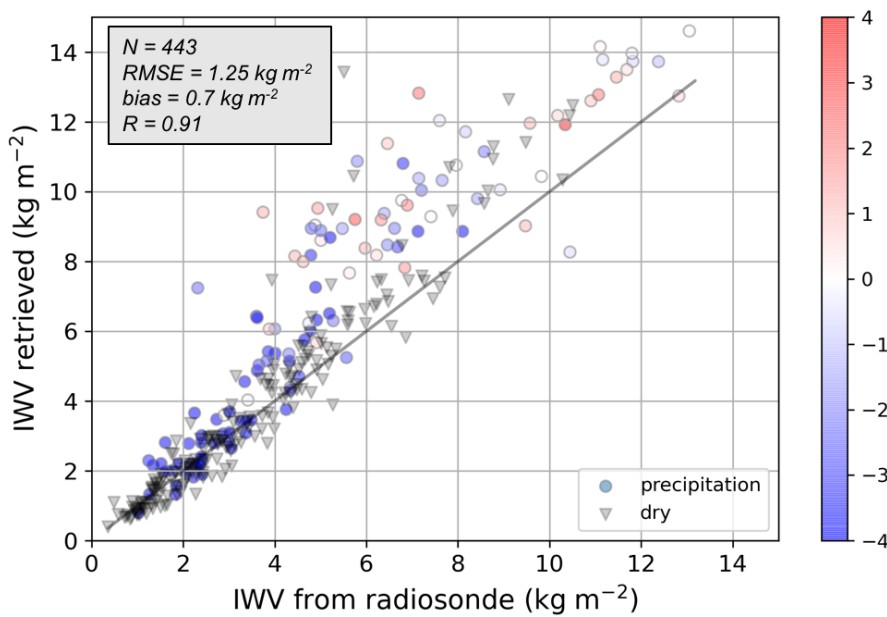

**Figure 13.** Scatter plot of retrieved IWV vs. IWV computed from radiosonde profiles. The algorithm used for the retrieval is the one with the full set of input features. The color indicates the surface temperature (in degrees Celsius). Dry conditions are identified with the equivalent radar reflectivity in the first kilometer above the radar (with a -10 dB threshold), and are coded as black triangles; precipitating conditions are denoted with circles. The size of the data set is indicated (N) as well as relevant error metrics (RMSE, mean bias, R).

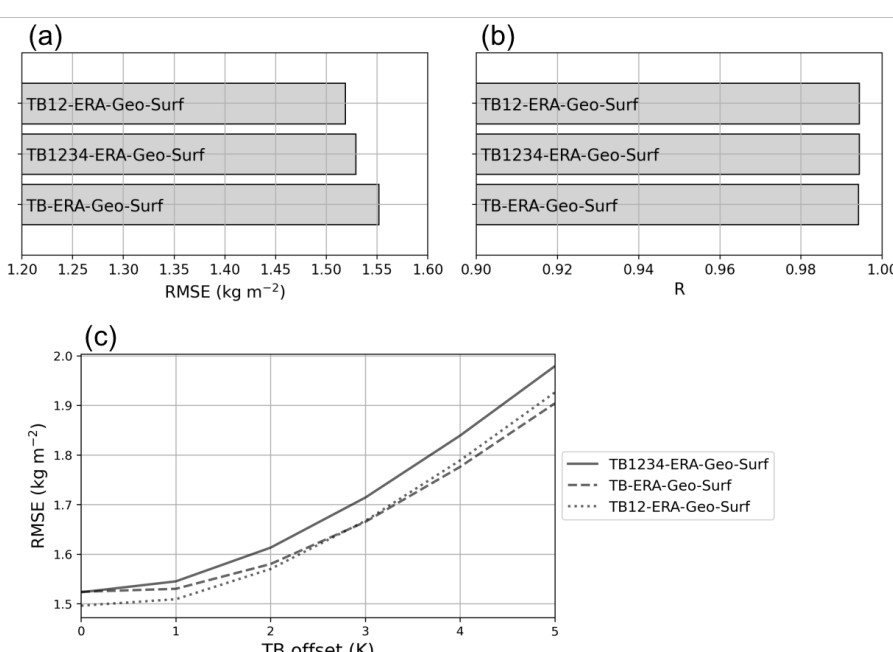

**Figure A1.** Effect of higher-order $T_B$ polynomials on IWV retrieval. Panels a) and b) show the RMSE and R on the testing set. It comes across that the best results are obtained with $T_B$ and $T_B^2$. Adding $T_B^3$ and $T_B^4$ leads to similar results, with slightly higher RMSE. Panel c) illustrates how the RMSE changes when a constant $T_B$ offset is added to the testing input, simulating a miscalibration of the radiometer. In terms of relative increase, the retrieval with $T_B$ only is slightly less affected, but it does not bring a large enough improvement to be considered preferable in comparison with the retrieval with $T_B$ and $T_B^2$.

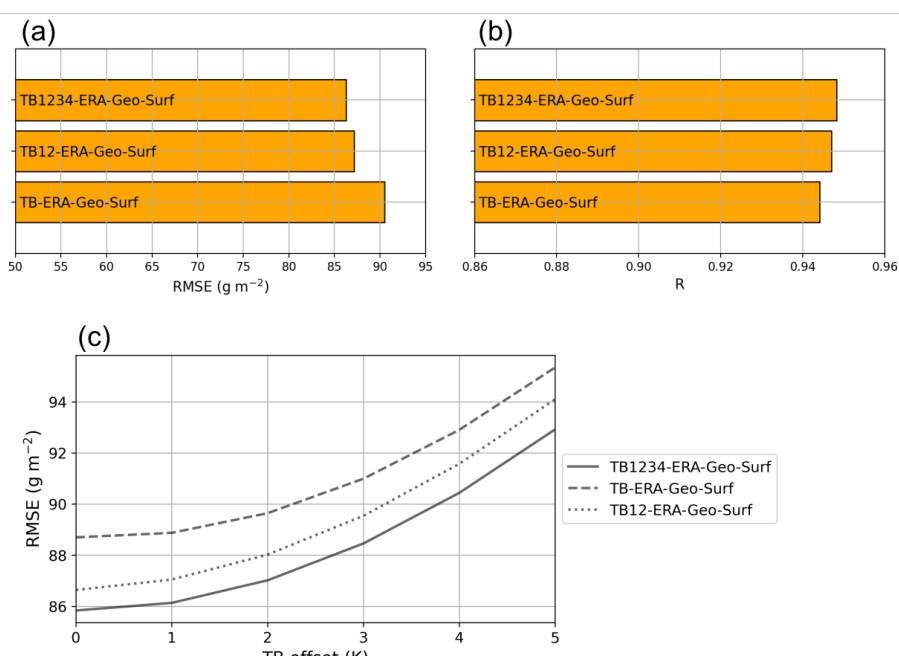

**Figure A2.** Effect of higher-order $T_B$ polynomials on LWP retrieval. Panels a) and b) show the RMSE and R on the testing set. It comes across that the best results are obtained with the full set of $T_B$ polynomials up to the fourth order. Using only $T_B$ and $T_B^2$ leads to similar results, with slightly higher RMSE. Panel c) illustrates how the RMSE changes when a constant $T_B$ offset is added to the testing input, simulating a miscalibration of the radiometer. In terms of relative increase, the retrieval with $T_B$ only is slightly less affected, but the RMSE remains higher than that of the other retrievals.

**Table 1.** Main parameters of the neural networks and training process.

| Target | Neurons | Layers | Cost function | Optimizer | Activation | Epochs | Batch size |
|--------|---------|--------|---------------|-----------|------------|--------|------------|
| IWV | 120 | 7 | Mean square error | RMS prop | ReLU | 70 | 512 |
| LWP | 150 | 6 | Mean square error | RMS prop | ReLU | 90 | 512 |

**Table A1.** Error metrics. X refers to LWP or IWV, and N is the length of the considered dataset.

| Root-mean-square error | Relative error | Bias | Correlation coeff. (R) |
|------------------------|----------------|------|------------------------|
| $[\frac{1}{N}\sum_{k=1}^{N}(X_{\mathrm{retrieved},k} - X_{\mathrm{target},k})^2]^{\frac{1}{2}}$ | $\frac{1}{N}\sum_{k=1}^{N}\frac{\lvert X_{\mathrm{retrieved},k} - X_{\mathrm{target},k}\rvert}{X_{\mathrm{target},k}}$ | $\frac{1}{N}\sum_{k=1}^{N}(X_{\mathrm{retrieved},k} - X_{\mathrm{target},k})$ | Pearson |

$\mathbf{X_{retrieved}}$ and $\mathbf{X_{target}}$ are length-N real positive vectors with the values of predicted (i.e. algorithm-retrieved) and target values, respectively. For the calculation of Relative error, $X_{\mathrm{target},k} = 0 \ \mathrm{g/m}^2$ are excluded from the dataset.