# Peer review of "Integrated water vapor and liquid water path retrieval using a single-channel radiometer"

_Atmospheric Measurement Techniques, 2020_

## Referee Comment (RC1) · Anonymous Referee #1 · 13 Sep 2020

General comments:

The authors developed an algorithm to retrieve liquid water path (LWP) and integrated water vapor (IWV) from ground-based passive microwave observations at 89 GHz. The algorithm is based on a neural network approach and uses synthetic cloud data derived from global radiosonde observations of atmospheric profiles. The algorithm is tested at two locations – Switzerland and South Korea, and the testing results are reasonably good. Because location and time information are used in the input of the algorithm, the algorithm can be used anywhere over the globe without further tuning required. The algorithm is targeted for the WProf radar-radiometer instrument, which

was deployed during the ICE-POP 2017 field experiment. I found that the work is quite useful, and the writing is generally clear. However, I do like to see some revisions before recommending publication. In particular, I'd like the authors to clearly state the fundamental limitation of the approach: the 89 GHz observation is not optimized for IWV retrieval, most of the IWV info gained from the algorithm is from secondary input parameters, such as surface temperature/humidity obs, ERA reanalysis, or even the correlation between LWP and IWV. I recommend minor revision, but would like to see the authors clearly address the above comment.

Specific comments:

Lines 9-11. The description of algorithm performance is too qualitative (don't carry much information). Please use some quantitative measures, such as R2, Bias, RMSE. Specify the performance separately for IWV and LWP retrievals. Also, indicate which input parameter(s) has (have) the most impact on the algorithm's performance.

Line 27. Aerosols contribute to millimeter wave radiation?

Line 40. I don't understand why physical model is "computationally heavy and cannot implemented accurately when only one frequency is available". I think the fundamental problem is that you have only one piece of information (TB), and yet you want to obtain two unknowns (IWV and LWP). You need secondary info and/or climatological relation between the unknowns. It is a highly statistical problem anyway; there is no need (not helpful) to go through the physical route.

Section 3.1: Please give more detailed info on how "clouds" are created based on atmospheric profiles, in particular, on how LWP depends on temperature and humidity. Since the observation is TB from one channel only, it cannot separate between liquid and vapor information. Therefore, if there is a correlation between humidity (or temperature) and cloud liquid, the retrieved LWP will be always correlated with IWV because the relation is built in the a priori training dataset. This correlation may not be valid in natural clouds, at least for some types of clouds. So, the authors should let readers

know about this issue, so that readers can take precaution when interpreting retrieved results. I suggest that the authors rewrite this section, provide full information on the liquid cloud model.

Lines 140-141. Higher order terms of TB are used as input of the algorithm. Why? Do the higher order terms actually have "the greatest importance"? I didn't see the importance has been shown in the sensitivity test section. Please clarify.

Line 145-146: I am concerned about the algorithm replying on reanalysis data, in particular directly replying on IWV and LWP in the reanalysis. This makes the "observation" completely mixed with "model", something worries me if I try to use "observation" to validate models.

Line 152: What are considered to be "strong rain events"? 0.1 mm/hr, 1 mm/hr, or 10 mm/hr? If I am a data user, I will stop using the retrieval when rainrate exceeds 0.1 mm/hr because all assumptions used in the algorithm development become invalid once rain/drizzle starts. So, I wouldn't call "strong rain" here.

Line 157. I am not sure I understand why if clear cases are left in the training set, "the training phase will result in a strong bias of the retrieval toward low LWP values." I thought the algorithm should be able to retrieve zero LWP (clear) cases. If all/most cases in the training set are cloudy cases, will the retrievals be most likely to have LWP>0? That is not what really happens in nature. For most locations, clear-sky has more frequency of occurrence than cloudy-sky. The fact that the case counts in Fig.2b is the same values for LWP from 0 to 600 g m^-2 also puzzles me. Maybe I misunderstand the concept here. Please explain.

Line 168. What is the difference between validation set and testing set?

Fig.6: Explanation to Fig.6 is severely lacking. It is really hard to understand what input are critical, although I guess the purpose of having this figure is exactly tended to show the importance of each input variable. Please think a better way to explain. But one

thing seems to be clear – no extra inputs but TB alone will result in a horrible retrieval. It also seems to me that ERA and the combination of Geo+Surf are redundant, i.e., you only need either, but doesn't have to have both, although the authors didn't conclude in such a way. If my interpretation is correct, I would get rid of ERA in the algorithm, and wouldn't say "the most important secondary feature is ERAs estimates." (lines 192-193).

Section 5.1. I would like to see a test in which you only use secondary info (without TB observation), and see how IWV and LWP "retrievals" will look like. I'd guess IWV retrievals could be reasonably good. If that is true, it means TB observation does not add much info to the retrievals.

Lines 245-247. Any ideas why error in these areas are large? Underrepresentation of radiosonde profiles in the training set, or cloud model not suitable for these regions?

Lines 261-262 "This . . .stage". This sentence doesn't make sense to me. ERA5 just happened to have IWV bias in this region for this time period, or ERA5 generally has IWV bias in general?

Fig.10. I have difficulties to interpret the results of the lowest two rolls in Figs.10c and d. Adding ERA data makes LWP results worse? I am confused here.

Lines 296-297. I always believed that the 89 GHz is not a good frequency for IWV. It works well for LWP. When IWV and LWP are not well correlated, IWV retrievals will suffer.

Lines 303-304. I am not sure that the results in this paper indicated this statement. Please explain.

Section 7. The summary appears mostly focusing on IWV retrieval. Please also mention some results on LWP. To repeat my main concern: I think this instrument is not good for IWV, but rather good for LWP. I hope that the authors can state something to reflect this point.

Technical comments:

Fig.2. The x-axis label in Fig.2b should be "LWP (g mˆ-2). Fig.3 and Table 1 are not explained well in the text. I am not sure what the curves and symbols mean. Please explain more as not everyone is familiar to the Keras library in Python.

———————————————

---

## Referee Comment (RC2) · Anonymous Referee #2 · 15 Sep 2020

AMT-2020-311: Integrated water vapor and liquid water path retrieval using a single-channel radiometer Anne-Claire Billault-Roux1 and Alexis Berne

The manuscript presents a single channel retrieval for IWV and LWP using the 89 GHz frequency. The retrieval uses a neural network trained with a synthetic dataset from a collection of radiosonde data worldwide. The coefficients are applied to the test dataset and to real data collected during two field campaigns. The retrievals provide robust, albeit not excellent results. The quality of the retrievals is however good and robust enough to be acceptable when more sophisticated instruments are not available. I found the paper interesting and the results useful considering the difficulty of deploying

full radiometric suites in many locations.

Overall the exposition is clear and well organized. I have some comments and questions that are listed below:

Line 140: "The first category consists of TB and higher order polynomials (up to fourth degree)" -Do higher order polynomials actually add information to the network? I would imagine that non-linearity is accounted for in the network structure.

Line 157: "In order to avoid this, the dataset was subsampled so that clear-sky and cloudy cases (up to 600 g m−2) would be equally represented;" -I am not sure I entirely agree with this approach. The point here is that neural networks perform better when the training dataset reproduces statistically the true occurrence of the events. Statistically clear sky cases occur more often than cloudy cases so I am afraid that modifying that distribution may actually cause the network to bias the LWP. Note that this is different from what you did earlier to avoid the uneven geographical sampling. In that case the problem was due to an uneven distribution of the monitoring network and the resampling was legitimate.

Section 5.1.1, line 193: Does the retrieval contribute anything to the ERA estimates of IWV? i.e. if you compute the RMSE of the ERA water vapor on the same dataset would you get the same RMSE as the retrievals (1.6 kg/m2) or worse? I see you show this information later on, but it would be useful to also comment on it here.

Section 5.1.2, line 210-2-13: "Similar reasons can help explain why the addition of reanalysis data significantly improves the IWV retrieval, but only in a minor way does it increase the LWP retrieval's accuracy. Liquid water content can vary on a shorter spatial and temporal scale than that captured by ERA5 models." -Another reason is that the 89 GHz is not a water vapor resonance therefore the information content for vapor is less than for liquid water path.

Figs 6, 10, 11, and 12 are a little bit difficult to interpret in my opinion. I am not sure I

understand them entirely. The general conclusion that I would draw from them is that the RMSE is very similar for almost all combinations of input parameters except for two or three combinations. However, looking at the various combinations, is hard to understand the rationale for the different performances. As an example if I look at Fig. 10a I see that the combination noERA-Geo-noSurf has a vastly different RMSE than noERA-noGeo-Surf. Does this mean that the effect of having surface parameters is equivalent to having ERA data? Similarly, for example in fig. 10b I see that the combination ERA-noPWVpred-Geo-Surf has much higher RMSE than ERA-noPWVpred-noGeo-noSurf. I am not sure how to interpret that.

---

## Referee Comment (RC3) · Anonymous Referee #3 · 22 Sep 2020

General Comments

1.) It's very good and important to see, that an approach has been made to derive a globally valid retrieval algorithm for IWV and LWP observations from single channel microwave observations. This can be beneficial for different science applications, e.g. weather & climate but also in astronomy and radio propagation.

2.) The paper is well written and makes clear points. It nicely addresses the fact, that the combination of microwave radiometry in synergy with re-analysis output and standard environmental conditions can be advantageous.

3.) Although I strongly favor short and concise papers, the results presented here

(especially in Sections 5 and 6) are – to an extent – kept rather minimal. The paper would benefit from a more detailed and quantitative discussion. See also my specific comments below.

Specific Comments

Section 1

Lines 25-27: If the authors are referring to TBs in the microwave spectrum, please omit "radiative contribution of . . . aerosols"

Section 2

Lines 68 onward: I assume you performed a quality control for the radiosonde profiles used for retrieval development, if not please consider doing so. Depict checks concerning range (e.g. min/max) of atmospheric parameters, maximum ascent height, consistency checks concerning pressure and/or temperature gradient, etc..

Line 76: Please discuss what the relatively low vertical resolution could imply for the retrieval development. E.g., the coarse resolution of the relative humidity profile will influence where and how many liquid layers are detected. What happens if this is significantly different for different radiosonde sites? This discussion could also be part of Section 3.1.

Lines 92/93: Describe how far WProf and HATPRO were apart (in meters) and if you expect any corresponding uncertainties during retrieval application.

Section 3

Section 3.2: I am missing a specification of the absorption models used for water vapor, oxygen and cloud liquid water. As previous studies have shown (e.g. Cimini et al. 2018, https://doi.org/10.5194/acp-18-15231-2018), this can be decisive for the absolute accuracy of your retrieval results. This aspect is nowhere discussed in the paper, should be, however.

Line 127: Please specify at which LWC (together with the assumed parameters of your gamma distribution, specify these as well), Mie effects become non-negligible and in how many cases in your data set this threshold is exceeded.

Section 4

Section 4.1: I don't find any indication in the manuscript of how you are dealing with random uncertainty of your measurement variables, e.g. radiometric noise and T/q/p sensor uncertainty. If you want to simulate a realistic retrieval behavior, you need to put noise on your measurements (training, validation and test data set), otherwise you are assuming a "perfect relationship" between measurement and LWP/IWV, which you will never have in reality.

Line 140: Since the relation ship between LWP/IWV and the TBs is, well I'd say, linear to moderately non-linear, I ask myself why you need to use $TB^4$? Can you quantify the retrieval improvement when using only TB and $TB^2$ in comparison to higher order terms?

Line 157: You mention "a strong bias of the retrieval toward low LWP values". Please quantify and compare to the retrieval with equally distributed LWP so you can justify this procedure.

Section 5

Begin with introducing your results in a general positive sense, make the reader feel like you are now going to present some great, interesting and relevant plots (which you mostly do!). I wouldn't begin Section 5.1 with two sentences that actually belong in the figure caption.

Section 5.1: I assume Figure 5 illustrates the retrieval including all input parameters? Please state this clearly in the text and figure caption.

Lines 188 – 190: I think Figure 5 would benefit from two additional sub figures showing the bias as a function of binned IWV and LWP. Then you could quantify the statement

you make in these two lines and elaborate a bit more on the bias behavior for smaller IWV.

Lines 191 – 194: I'm a bit puzzled by Figure 6a. If you only use the 89 GHz TB then ERA5 performs significantly better. So, is there any sense of using this TB at all? I think you need to perform a retrieval without TB, just with all other parameters and add this one to your plots. This would help in putting the value of the 89 GHz TB in context. Then you need to discuss your results in more detail.

Lines 196 and following: When you mention the RMSE of 86 $gm^{-2}$, I assume you are applying the retrieval derived from the equally LWP-distributed training data set to the equally distributed test data set? I'm not sure.. please make clear.

Lines 199 – 201: You write: "with however a bias for low LWP values, which are slightly overestimated, and for large LWP ($> 800$ g m$-2$) which are underestimated". Please apply my comment to Lines 188 – 190.

Line 211: You write: "but only in a minor way does it increase the LWP retrieval's accuracy". But it does!? Going from noERA-noIWVpred-noGeo-noSurf to ERA-noIWVpred-noGeo-noSurf reduces the RMSE from roughly 140 to 90 $gm^{-2}$. Or am I misinterpreting something wrong here?

Section 5.2: I like the idea of showing the sensitivity to instrument calibration offset, but only when I look at Figure 7, do I only see that you have looked at the effects for all different retrieval configurations and for continuously rising TB offset. Here again: please describe and discuss your results with more detail. To make your discussion complete, please add the "only-TB" retrieval to Figure 7a and 7b.

Section 5.3: Do you have any interpretation as of why the results over the Indian Peninsula are so much worse than elsewhere, even compared with sites at similar latitude? Is it possible that this is associated to the quality of the radiosondes or is there any other reason you can think of?

Line 242: Please describe how you think "humidity and temperature conditions" can lead to the discrepancy.

Figure 9: Can you explain the outliers (especially the vertical and horizontal "bar structures") in Figures 9a and 9b?

Section 6

I'm missing a discussion of Figure 11. One point would be, e.g. as also seen in Figure 6a, that the reanalysis performs better than the TB-only retrieval. Here, again it would help if you added a retrieval derived without any TB to discuss the overall TB value. Comparing such a retrieval against your ERA-Geo-Surf would tell you something about the impact of 89 GHz TB and if it's sensible to use it at all if you have the other parameters available.

Line 261: Please quantify the "constant bias"

Lines 296 and following: In Fig 12, I'd also include results from a retrieval without TB. Another possibility for ERA5 outperforming the retrievals could maybe be fog? Could you include a discussion of the weather during ICE-POP?

Section 7

Can you elaborate a little on what the alternative would be to using the re-analysis? If, e.g. you would need quasi real-time retrieval results.

Figure 2

X-axis labelling of Figure 2b needs to read "LWP", not "IWV"

Figure 6

Do you need to make the bars orange with diagonal lines – just orange would probably make the text easier to read?

Figure 9

It would help if you included in-plot statistics such as number of cases, bias, RMSE and $R^2$. Best do consistently with Figures 5 and 13.

Figure 10

I think the ordering of the text in the bars is swapped, otherwise the plots make no sense to me.

---

## Author Comment (AC1) · 7 Dec 2020

We thank the reviewer for their comment. The detailed response is provided in the attachment.

Please also note the supplement to this comment:
https://amt.copernicus.org/preprints/amt-2020-311/amt-2020-311-AC1-supplement.pdf

---

## Author Comment (AC2) · 7 Dec 2020

**Integrated water vapor and liquid water path retrieval using a single-channel radiometer**

**amt-2020-211**

**Responses to reviewers**

Anne-Claire Billault-Roux and Alexis Berne

November 2020

First, we would like to thank the reviewers for their constructive comments that helped us improve our retrieval method and perform a more thorough analysis of its quality. In the present document, we provide our responses to the comments of the three referees. Each section corresponds to the comments of a referee. The comments of the reviewers are reported in italic, our responses in normal font and the corresponding modifications in the manuscript in blue.

In addition to the modifications related to referee comments, other minor changes were made to the manuscript, which are listed below:

**Section 3.2** *"The cloud droplet size distribution (DSD) is chosen as a gamma distribution following Karstens et al. (1994)"*
The cloud droplet size distribution (DSD) is chosen as a monodisperse distribution with radius rc = 20 $\mu m$ following Cadeddu et al. (2017)

It was considered preferable to use a unique cloud droplet size distribution, rather than one that introduces empirical thresholds. As explained in Sect. 3.2, the exact choice of the DSD does not affect the radiative transfer model as long as droplets are within the Rayleigh range for the considered frequency. This corresponds to the range of validity of the algorithm, so this modification does not substantially affect the results.

**Section 6.2** *"The value of this offset (18 K) was determined by computing theoretical brightness temperatures from clear-sky radiosonde profiles"*
The value of this offset (20 K) was determined by computing theoretical brightness temperatures from clear-sky radiosonde profiles

Calculations to evaluate the miscalibration were performed once again. The exclusion of precipitating cases requires to define some thresholds, which were adjusted during this new analysis. This led us to modify the value of the offset in TB. However, this value should be taken with care, since a fully satisfactory correction of such a high miscalibration is a quite challenging task.

**1    Anonymous referee #1**

*The authors developed an algorithm to retrieve liquid water path (LWP) and integrated water vapor (IWV) from ground-based passive microwave observations at 89 GHz. The algorithm is based on a neural network approach and uses synthetic cloud data derived from global radiosonde observations of atmospheric profiles. The algorithm is tested at two locations – Switzerland and South Korea, and the testing results are reasonably good. Because location and time information are used in the input of the algorithm, the algorithm can be used anywhere over the globe without further tuning required. The algorithm is targeted for the WProf radar-radiometer instrument, which was deployed during the ICE-POP 2017 field experiment. I found that the work is quite useful, and the writing is generally clear. However, I do like to see some revisions before recommending publication. In particular, I'd like the authors to clearly state the fundamental limitation of the approach: the 89 GHz observation is not optimized for IWV retrieval, most of the IWV info gained from the algorithm is from secondary input parameters, such as surface temperature/humidity obs, ERA reanalysis, or even the correlation between LWP and IWV. I recommend minor revision, but would like to see the authors clearly address the above comment.*

We thank the reviewer for the positive comment and the constructive feedback. This raises a valid point that we did not emphasize in the first version of the manuscript: the frequency of the microwave radiometer which is used here is mostly sensitive to liquid water, and to a smaller extent to water vapor. Initially, the algorithm was only designed for LWP retrieval, but in the course of investigations it was found that it could also provide reasonable estimates of IWV, provided that other input features are taken into account. We tried to make this more clear in the revised version, for example through the following items.

[Abstract] While 89-GHz brightness temperature is crucial to LWP retrieval, only moderately does it contribute to IWV estimation, because of the lower sensitivity of this channel to water vapor emission.

[Section 1] Those additional input features are especially key to the retrieval of IWV, to which TB at 89 GHz is less sensitive than to LWP.

[Section 5.1.1] If only one input feature were available, all the versions would predict worse results than those given by reanalysis data. Including TB in the retrieval does not lead to the same leap in accuracy than for LWP (discussed in the following subsection), which was expected because the microwave frequency that is used here is not highly sensitive to water vapor. However, excluding TB from the input features degrades the RMSE to 2.56 kg m$^{-2}$, i.e. + 67 % error compared to the best version, which clearly shows that some information is extracted from the brightness temperature in the retrieval.

[Section 5.1.2] This also highlights once again that brightness temperature at 89 GHz is much more sensitive to liquid water than to water vapor: for LWP retrieval, input features other than TB only bring second-order improvements, while they were shown to be crucial in the IWV retrieval.

[Section 6.2] Because of the relatively low sensitivity of TB at 89 GHz to water vapor, the algorithm largely relies on non-radiometric features; this is even more the case in cold and dry environments like that of ICEPOP, where IWV is low.

[Section 7] While retrieving IWV based on $T_B$ at 89 GHz alone does not lead to accurate results – because of the lower sensitivity of this channel to water vapor emission, compared to

liquid water – this study showed that reliable retrievals could be achieved by including surface and geographical information, as well as reanalysis data if available, among the input features.
* * *
1. *Lines 9-11. The description of algorithm performance is too qualitative (don't carry much information). Please use some quantitative measures, such as R2, Bias, RMSE. Specify the performance separately for IWV and LWP retrievals. Also, indicate which input parameter(s) has (have) the most impact on the algorithm's performance.*

We added some detail on the algorithm's performance in the abstract. As such, we believe it might be misleading to cite in the abstract which of the secondary input features has the greatest effect, since we demonstrate that they each contribute to the retrieval in a relatively similar way. The take-home message is that overall, including diverse features (rather than using TB alone) improves significantly the algorithm.

The new algorithm is shown to be quite robust although its accuracy is inevitably lower than that obtained with state-of-the-art multi-channel radiometers, with a relative error of 7.2 % for LWP (on cloudy cases with LWP > 30 g m$^{-2}$) and 5.2 % on IWV. The highest accuracy is obtained in mid-latitude environments with a moderately moist climate. Furthermore, the additional input features are found to improve significantly the accuracy of the retrieval.
* * *
2. *Line 27. Aerosols contribute to millimeter wave radiation?*

We thank the reviewer for noting this mistake, which we removed.
* * *
3. *Line 40. I don't understand why physical model is "computationally heavy and cannot implemented accurately when only one frequency is available". I think the fundamental problem is that you have only one piece of information (TB), and yet you want to obtain two unknowns (IWV and LWP). You need secondary info and/or climatological relation between the unknowns. It is a highly statistical problem anyway; there is no need (not helpful) to go through the physical route.*

Physical methods are used in some state-of-the-art multi-frequency radiometer retrievals, as described for example in Cadeddu et al. 2013 and Turner et al. 2007. In those cases, such methods are more accurate than statistical retrievals, but require more initial information on the atmospheric column; in addition, they have a higher computational expense. As highlighted by the referee in this comment, it is in our case not possible to go through such a physical method because of the lack of microwave measurements. Thus, it was misleading to mention the high computational expense - which was in this case not the problem.

Although this method is formally the most accurate (Turner et al. 2007), it requires more than one radiometer frequency to lift the problem's fundamental underdetermination, and is thus not applicable for this study

4. *Section 3.1: Please give more detailed info on how "clouds" are created based on atmospheric profiles, in particular, on how LWP depends on temperature and humidity. Since the observation is TB from one channel only, it cannot separate between liquid and vapor information. Therefore, if there is a correlation between humidity (or temperature) and cloud liquid, the retrieved LWP will be always correlated with IWV because the relation is built in the a priori training dataset. This correlation may not be valid in natural clouds, at least for some types of clouds. So, the authors should let readers know about this issue, so that readers can take precaution when interpreting retrieved results. I suggest that the authors rewrite this section, provide full information on the liquid cloud model.*

More detail was included on the liquid cloud model in Sect. 3.1. It is indeed true that the model might not be accurate in certain cases, and can fail to output the correct LWP for a given atmospheric profile. However, the most important here is that the model is bias-free and outputs a *realistic* – if not *true* – profile, which is then used to create the dataset and to train the retrieval algorithm. The Salonen model was compared to a few commonly used models (described for example in Mattioli 2009, Loehnert and Crewell 2003), and selected because it provided the least mean bias when compared to ERA5 data. This ensured that the model is realistic enough to produce a dataset on which to perform the statistical learning.

To derive profiles of liquid water content (LWC) from radiosonde profiles of atmospheric variables, the cloud model from Salonen 1991 was used. Cloud boundaries are identified using a threshold $U_c$ on relative humidity, this threshold being pressure- and temperature-dependent, according to Eq. 1.

$$U_c = 1 - \alpha\sigma(1 - \sigma)[1 + \beta(\sigma - 0.5)] \tag{1}$$

Here, $\sigma = \frac{P}{P_0}$ with $P$ and $P_0$ denoting respectively atmospheric pressure at the current level and at the ground. Corrections from Mattioli et al. 2009 are used for the coefficients $\alpha$ and $\beta$ of the Salonen model. Within the cloud layers, the liquid water profile is then calculated as a function of temperature and height above cloud base, following Eq. 2.

$$LWC = w_0(\frac{h - h_b}{h_r})^a f(T) \tag{2}$$

where $f(T) = 1 + cT$ for $T \geq 0$ and $f(T) = exp(cT)$ for $T < 0$, with $T$ in °C, $c = 0.04°C^{-1}$, $w_0 = 0.17$ g m$^{-3}$, $h_r = 1.5$ km, $h$ and $h_b$ denoting height and height of cloud base. There are some limitations when assuming a single universal cloud model, since it may fail to capture specific cloud properties in certain environments: more sophisticated and accurate models can be defined on a local geographical scale (e.g. Pierdicca et al. 2006). However, given the stated objective of this study to design a non-site specific algorithm, it was considered preferable to assume a single universal liquid cloud model, in spite of its potential drawbacks.
A further limitation of this cloud model is related to the relatively low resolution of the atmospheric profiles extracted from the radiosonde data (c.f. Sect. 2.1) that are used as an input. This might result in a misrepresentation of the cloud layers in their detection and their size.
In order to ensure that this forward model resulted in the least possible bias, its results were compared against LWP values from reanalysis data. Even though the model might fail, on a given occurrence, to reproduce the actual liquid water profile in the atmospheric column, it should not produce a significant bias on average. This condition guarantees that the synthetic dataset that is used for training contains realistic – if not real – profiles, and this should therefore not degrade the quality of the retrieval algorithm. This cloud model was chosen over other commonly used ones (Decker model, Salonen model without correction, c.f. Mattioli 2009) for

it was found to produce the least bias when compared to ERA5 LWP values (mean bias of 14 $g\,m^{-2}$ vs. 26 $g\,m^{-2}$ (resp. -24 $g\,m^{-2}$) for the unadjusted Salonen model (resp. the Decker model with 95 % threshold).
* * *
5. *Lines 140-141. Higher order terms of TB are used as input of the algorithm. Why? Do the higher order terms actually have "the greatest importance"? I didn't see the importance has been shown in the sensitivity test section. Please clarify.*

Indeed, an analysis of the importance of higher order terms was missing from the first version of the manuscript and was included in Sect. 5. Figures to support this analysis are presented in Appendix.

[Section 5.1.1] An analysis was conducted to identify the importance of higher order polynomials in the algorithm, a summary of which can be found in Appendix. It was found that the most accurate retrieval is obtained by including TB and $TB^2$. If higher order terms are added, this slightly reduces the accuracy of the retrieval, and also degrades its stability to TB miscalibration. On the other hand, including only TB, while it makes the algorithm slightly more stable, does not appear as the best solution for it has lower accuracy. Hence, the results which are presented here and in the further sections are those obtained using TB and $TB^2$.

[Section 5.1.2] The analysis of higher order terms' importance in the case of LWP retrieval shows that the best results are obtained by using TB polynomials up to the fourth order (cf. Appendix), while this does not affect significantly the stability of the retrieval to errors in TB. Let us highlight that in the case of a linear regression, one would expect the error to diverge when high-order polynomials are included. This is not the case here, because of the non-linear behavior of the neural network. Therefore, in the results which are shown here and further, "TB" implies that TB polynomials up to the fourth order are used.
* * *
6. *Line 145-146: I am concerned about the algorithm replying on reanalysis data, in particular directly replying on IWV and LWP in the reanalysis. This makes the 'observation" completely mixed with "model", something worries me if I try to use "observation" to validate models.*

The logic behind this point is that reanalysis data can contain information on the meteorological and geographical setting of a given measurement. The spatial and temporal resolution is not sufficient for the reanalysis to be considered as ground truth, and is therefore not used for the validation of the algorithm. As such, ERA5 data is not incorporated in a physical model, but merely serves to improve the statistical retrieval.

The spatial and temporal resolution of this reanalysis data is too low for it to be held as ground truth, but it can serve as a reasonable rough estimate and thus bring some improvements to the statistical learning process – although it could not be included as such in a physical model.
* * *
7. *Line 152: What are considered to be "strong rain events"? 0.1 mm/hr, 1 mm/hr, or 10 mm/hr? If I am a data user, I will stop using the retrieval when rainrate exceeds 0.1 mm/hr because all assumptions used in the algorithm development become invalid once rain/drizzle starts. So, I wouldn't call "strong rain" here.*

We changed "strong rain" to "rain" and included a more detailed discussion of this topic in Sect. 3.2, in relation with one comment from Referee #3.
* * *
8. *Line 157. I am not sure I understand why if clear cases are left in the training set, "the training phase will result in a strong bias of the retrieval toward low LWP values." I thought the algorithm should be able to retrieve zero LWP (clear) cases. If all/most cases in the training set are cloudy cases, will the retrievals be most likely to have LWP >0 ? That is not what really happens in nature. For most locations, clear-sky has more frequency of occurrence than cloudy-sky. The fact that the case counts in Fig.2b is the same values for LWP from 0 to 600 g m$^{-2}$ also puzzles me. Maybe I misunderstand the concept here. Please explain.*

If the training set is kept as such – without subsampling –, the algorithm has a huge bias toward low values: because the training set contains almost only values of LWP=0, the algorithm has very good performance by predicting LWP=0 all the time. Which is not what is desired... This is an example of regression toward the mean, a common behavior for regression methods. To correct for this, we truncated the distribution. This does not mean excluding clear-sky cases (they are still present in the training set, although in a proportion more comparable to cloudy cases), and the algorithm is still able to identify LWP=0. The choice of the 600 g/m2 threshold resulted from a trial-error method. We modified the explanation to clarify this point.

If left as such, the training phase would result in a strong bias of the retrieval toward low LWP values (a bias of $\sim 100$g m$^{-2}$ for LWP $> 400$g m$^{-2}$ was noted): this is a common artefact that is observed in statistical learning algorithms, as an effect of unbalanced training set. In order to avoid this, the dataset was subsampled so that clear-sky and cloudy cases (up to 600 g m$^{-2}$) would be equally represented; the value chosen for this threshold results from a trade-off between bias reduction and preservation of overall accuracy.
* * *
9. *Line 168. What is the difference between validation set and testing set?*

The hyperparameters of the neural network (number of neurons, layers...) are tuned on the validation set while the testing set is for the final evaluation of the algorithm.

The validation set is used for tuning the hyperparameters of the neural network, while the final evaluation metrics are computed on the testing set.
* * *
10. *Fig.6: Explanation to Fig.6 is severely lacking. It is really hard to understand what input are critical, although I guess the purpose of having this figure is exactly tended to show the importance of each input variable. Please think a better way to explain. But one thing seems to be clear: no extra inputs but TB alone will result in a horrible retrieval. It also seems to me that ERA and the combination of Geo+Surf are redundant, i.e., you only need either, but doesn't have to have both, although the authors didn't conclude in such a way. If my interpretation is correct, I would get rid of ERA in the algorithm, and wouldn't say "the most important secondary feature is ERAs estimates." (lines 192-193).*

We tried to make the figure more readable by changing the labels and only writing the input variables which are used for each version of the algorithm. We also expanded the explanation and analysis of this figure in Section 5.1 which was largely rewritten.

[Section 5.1] Figure 6 shows how this total error, represented by RMSE (left panels) and by the square correlation coefficient ($R^2$) (right panels), is affected by the addition or removal of input features. For each set of input features, a full tuning of the algorithm was performed, and the results that are presented correspond to those from the tuned – i.e. best – version on the testing set.

[Section 5.1.1] From Fig. 6 a) and b) it comes across that the IWV retrieval is significantly improved by the addition of multiple input features. The highest accuracy is obtained with the full set of input features, and corresponds to a RMSE of 1.53 $kg\ m^{-2}$. On the other hand, including solely $T_B$ measurements in the input deteriorates the RMSE to nearly 6 $kg\ m^{-2}$. If only one input feature were available, all the versions would predict worse results than those given by reanalysis data. Including $T_B$ in the retrieval does not lead to the same leap in accuracy than for LWP (discussed in the following subsection), which was expected because the microwave frequency that is used here (89 GHz) is much less sensitive to water vapor. However, excluding $T_B$ from the input features degrades the RMSE to 2.56 $kg\ m^{-2}$,i.e. + 67 % error compared to the best version, which clearly shows that some information is extracted from the brightness temperature in the retrieval.

[Section 5.1.2] Figures 6 c) and d) show that for LWP retrieval, input features other than $T_B$ only bring second-order improvements, while they were shown to be crucial in the IWV retrieval. For instance, the addition of reanalysis data significantly improves the IWV retrieval, but only in a relatively minor way does it increase the accuracy of LWP retrieval. On the contrary, excluding $T_B$ from the input features leads to RMSE near 200 $g\ m^{-2}$ and $R^2 < 0.5$, i.e. to values with make the retrieval not relevant. This highlights once again that brightness temperature at 89 GHz is much more sensitive to liquid water than to water vapor. An additional reason for this high dependence on $T_B$ is that LWP at a given location can have a high temporal variability due to cloud dynamics in the atmospheric column, which might not always be captured in the time series of surface atmospheric variables, nor by ERA5 models which have a comparatively low spatial and temporal resolution.

11. *Section 5.1. I would like to see a test in which you only use secondary info (without TB observation), and see how IWV and LWP "retrievals" will look like. I'd guess IWV retrievals could be reasonably good. If that is true, it means TB observation does not add much info to the retrievals.*

This is an important point that we did not discuss in the first version of the manuscript. In the

revised version, a discussion of this point was included in Sect. 5.

[Section 5.1.1] Including $T_B$ in the retrieval does not lead to the same leap in accuracy than for LWP (discussed in the following subsection), which was expected because the microwave frequency that is used here (89 GHz) is much less sensitive to water vapor. However, excluding $T_B$ from the input features degrades the RMSE to 2.56 $kg\ m^{-2}$, i.e. $+\ 67\ \%$ error compared to the best version, which clearly shows that some information is extracted from the brightness temperature in the retrieval.

[Section 5.1.2] On the contrary, excluding $T_B$ from the input features leads to RMSE near 200 $g\ m^{-2}$ and $R^2 < 0.5$, i.e. to values with make the retrieval not relevant.
* * *
12. *Lines 245-247. Any ideas why error in these areas are large? Underrepresentation of radiosonde profiles in the training set, or cloud model not suitable for these regions?*

It is difficult to identify the most important factor that causes a high error in these areas. The underrepresentation of this range of latitudes and corresponding climates is certainly a critical point: unfortunately, there is to our knowledge little data available from these areas.
We would expect the cloud model to also be less accurate in such areas than in the midlatitude regions for which it was initially designed. However, this should not transpose directly to such a high error in the results of the algorithm on the synthetic dataset (which does not consist of "true" LWP but of LWP derived from this cloud model; so the bias from the cloud model should not be visible).

Tropical climates are underrepresented in the dataset, for little data is available from this region in comparison with mid-latitude areas: their specificity might therefore not be fully captured during the learning stage of the algorithm. This accounts at least partly for the enhanced error over the Indian peninsula and South-Eastern Asian islands.
* * *
13. *Lines 261-262 "This … stage". This sentence doesn't make sense to me. ERA5 just happened to have IWV bias in this region for this time period, or ERA5 generally has IWV bias in general?*

It is a bias that was noted by MeteoSwiss at Payerne, at least during this timeframe. This is not a general statement on ERA5 values. We tried to make this sentence clearer.

This might be due to a bias in ERA5 data during this timeframe over the region, which is visible in ERA5 records during the entire campaign (with a value of $-4.1 kg\ m^{-}2$) and for which there is no clear explanation at this stage.
* * *
14. *Fig.10. I have difficulties to interpret the results of the lowest two rolls in Figs.10c and. Adding ERA data makes LWP results worse? I am confused here.*

There was unfortunately a major mistake in the labels of this figure. We corrected for this, please refer to the corresponding Fig. 10 in the revised version.
* * *
15. *Lines 296-297. I always believed that the 89 GHz is not a good frequency for IWV. It works well for LWP. When IWV and LWP are not well correlated, IWV retrievals will suffer.*

Indeed, TB at 89 GHz is much more sensitive to liquid water than to water vapor. However, in general, removing TB from the input features still leads to a decrease in the accuracy of the algorithm, although in a much smaller way than for LWP, as discussed in Sect. 5.

The best results are found when several input features are included and drop severely when no secondary input features are used, which corresponds to the results on the synthetic data set presented in Sect. 5. Because of the relatively low sensitivity of TB at 89 GHz to water vapor, the algorithm largely relies on non-radiometric features; this is even more the case in cold and dry environments like that of ICEPOP, where IWV is low.
* * *
16. *Lines 303-304. I am not sure that the results in this paper indicated this statement. Please explain.*

This is an observation deduced from Fig. 13. The sentence was rephrased to avoid confusion.

Snowfall events during the campaign, as well as occasional fog, can also bias the retrieval by enhancing brightness temperature.
The analysis of the ICE-POP data was taken a step further to explore the latter point. It appears that the IWV retrieval is most reliable in non-precipitating or cold conditions, i.e. when little liquid water is expected in the column. To visualize this, periods with no precipitation are identified using WProf's radar measurements as time steps with low radar equivalent reflectivity ($Ze < -10dBZ$) in the lower gates (first kilometer above the radar), and temperature time series are provided by the weather station coupled to WProf. Fig. 13 shows the scatter plot of the error, color-coded in a way to differentiate dry conditions from precipitating cases.
* * *
17. *Section 7. The summary appears mostly focusing on IWV retrieval. Please also mention some results on LWP. To repeat my main concern: I think this instrument is not good for IWV, but rather good for LWP. I hope that the authors can state something to reflect this point.*

We nuanced the results on IWV and highlighted that the retrieval of IWV is a corollary to that of LWP, which we did not make clear in the original manuscript. For a reasonably accurate IWV retrieval, it is critical to have several ground measurements available; TB has a role but certainly not as crucial as in the LWP retrieval.

While retrieving IWV based on $T_B$ at 89 GHz alone does not lead to accurate results – because of the lower sensitivity of this channel to water vapor emission, compared to liquid water – this study showed that reliable retrievals could be achieved by including surface and geographical information, as well as reanalysis data if available, among the input features.

18. *Fig.2. The x-axis label in Fig.2b should be "LWP (g mˆ-2). Fig.3 and Table 1 are not explained well in the text. I am not sure what the curves and symbols mean. Please explain more as not everyone is familiar to the Keras library in Python.*

The label of Fig. 2 was corrected.
Some more detail was included on the design and training of the algorithm in section 4.3, to explain Fig. 3 and make more clear the notations of Table 1.

The neural network was trained through mini-batch gradient descent, using RMSprop optimizer which allows for learning rate adaptation and is often used for statistical regression problems (Chollet, 2017). As comes across from the training curve of the LWP retrieval on Fig. 3, the training dataset is large enough to ensure that the algorithm is not prone to overfitting. Indeed, the error on the validation set quickly drops when the size of the training set, then plateaus with a slight decrease. In other words, the accuracy of the algorithm is not limited by the amount of data used in the training stage. Figure 4 and Table 1 summarize the resulting architecture and relevant parameters of the algorithm. These include the description of the neural network's structure (number of neurons and hidden layers) as well as training parameters such as the batch size and number of epochs, i.e. the number of iterations through the entire dataset in the learning phase.

**2 Anonymous referee #2**

*The manuscript presents a single channel retrieval for IWV and LWP using the 89 GHz frequency. The retrieval uses a neural network trained with a synthetic dataset from a collection of radiosonde data worldwide. The coefficients are applied to the test dataset and to real data collected during two field campaigns. The retrievals provide robust, albeit not excellent results. The quality of the retrievals is however good and robust enough to be acceptable when more sophisticated instruments are not available. I found the paper interesting and the results useful considering the difficulty of deploying full radiometric suites in many locations. Overall the exposition is clear and well organized.*

We thank the reviewer for their positive comment.
* * *
1. *Line 140: "The first category consists of TB and higher order polynomials (up to fourth degree)" -Do higher order polynomials actually add information to the network? I would imagine that non-linearity is accounted for in the network structure.*

   While neural networks are able to fit non-linear behaviors, they are not always able to resolve polynomial features – or require more parameters to do so, with the risk of producing overfitting. Polynomial input features are thus sometimes used in the design of NNs. The concern of how much those higher order powers benefit to the retrieval is fully valid, and we included a more in-depth discussion of this topic in Sect. 5. Figures to support this analysis are presented in Appendix.

   [Section 5.1.1] An analysis was conducted to identify the importance of higher order polynomials in the algorithm, a summary of which can be found in Appendix. It was found that the most accurate retrieval is obtained by including TB and $TB^2$. If higher order terms are added, this slightly reduces the accuracy of the retrieval, and also degrades its stability to TB miscalibration. On the other hand, including only TB, while it makes the algorithm slightly more stable, does not appear as the best solution for it has lower accuracy. Hence, the results which are presented here and in the further sections are those obtained using TB and $TB^2$.

   [Section 5.1.2] The analysis of higher order terms' importance in the case of LWP retrieval shows that the best results are obtained by using TB polynomials up to the fourth order (cf. Appendix), while this does not affect significantly the stability of the retrieval to errors in TB. Let us highlight that in the case of a linear regression, one would expect the error to diverge when high-order polynomials are included. This is not the case here, because of the non-linear behavior of the neural network. Therefore, in the results which are shown here and further, "TB" implies that TB polynomials up to the fourth order are used.
* * *
2. *Line 157: "In order to avoid this, the dataset was subsampled so that clear-sky and cloudy cases (up to 600g $m^{-2}$) would be equally represented" -I am not sure I entirely agree with this approach. The point here is that neural networks perform better when the training dataset reproduces statistically the true occurrence of the events. Statistically clear sky cases occur more often than cloudy cases so I am afraid that modifying that distribution may actually cause the*

*network to bias the LWP. Note that this is different from what you did earlier to avoid the uneven geographical sampling. In that case the problem was due to an uneven distribution of the monitoring network and the resampling was legitimate.*

If the training set is kept as such – without subsampling –, the algorithm has a huge bias toward low values: because the training set contains almost only values of LWP=0, the algorithm has very good performance by predicting LWP=0 all the time. Which is not what is desired... This is an example of regression toward the mean, a common behavior for regression methods. To correct for this, we truncated the distribution. This does not mean excluding clear-sky cases (they are still present in the training set, although in a smaller proportion), and the algorithm is still able to identify LWP=0. The choice of the 600 g/m2 threshold resulted from a trial-error method. We modified the explanation to clarify this point.

If left as such, the training phase would result in a strong bias of the retrieval toward low LWP values (a bias of $\sim 100\text{g m}^{-2}$ for LWP $> 400\text{g m}^{-2}$ was noted): this is a common artefact that is observed in statistical learning algorithms, as an effect of unbalanced training set. In order to avoid this, the dataset was subsampled so that clear-sky and cloudy cases (up to 600 g m$^{-2}$) would be equally represented; the value chosen for this threshold results from a trade-off between bias reduction and preservation of overall accuracy.
* * *
3. *Section 5.1.1, line 193: Does the retrieval contribute anything to the ERA estimates of IWV? i.e. if you compute the RMSE of the ERA water vapor on the same dataset would you get the same RMSE as the retrievals (1.6 kg/m2) or worse? I see you show this information later on, but it would be useful to also comment on it here.*

The information was added in this section.

For comparison, the ERA5 data alone has a higher RMSE (3.4 kg m$^{-2}$) on the same data set.
* * *
4. *Section 5.1.2, line 210-2-13: "Similar reasons can help explain why the addition of reanalysis data significantly improves the IWV retrieval, but only in a minor way does it increase the LWP retrieval's accuracy. Liquid water content can vary on a shorter spatial and temporal scale than that captured by ERA5 models." -Another reason is that the 89 GHz is not a water vapor resonance therefore the information content for vapor is less than for liquid water path.*

This is an important aspect which was also pointed out by the other referees, and which we did not emphasize enough in the first version of the manuscript. We mention it here and also recall it in the summary section.

This also highlights once again that brightness temperature at 89 GHz is much more sensitive to liquid water than to water vapor: for LWP retrieval, input features other than TB only bring second-order improvements, while they were shown to be crucial in the IWV retrieval.
* * *
5. *Figs 6, 10, 11, and 12 are a little bit difficult to interpret in my opinion. I am not sure I understand them entirely. The general conclusion that I would draw from them is that the RMSE is very similar for almost all combinations of input parameters except for two or three combinations. However, looking at the various combinations, is hard to understand the rationale for the different performances. As an example if I look at Fig. 10a I see that the combination noERA-Geo-noSurf has a vastly different RMSE than noERAnoGeo-Surf. Does this mean that the effect of having surface parameters is equivalent to having ERA data? Similarly, for example in fig. 10b I see that the combination ERAnoPWVpred- Geo-Surf has much higher RMSE than ERA-noPWVpred-noGeo-noSurf. I am not sure how to interpret that.*

We tried to make these error plots more readable by changing the labels and only writing the input variables which are used for each version of the algorithm. We also expanded the explanation and analysis of figure 6 in Section 5.1 which was largely rewritten. There was unfortunately a major mistake in the labels of Fig. 10, which was corrected for.

[Section 5.1] Figure 6 shows how this total error, represented by RMSE (left panels) and by the square correlation coefficient ($R^2$) (right panels), is affected by the addition or removal of input features. For each set of input features, a full tuning of the algorithm was performed, and the results that are presented correspond to those from the tuned – i.e. best – version on the testing set.

[Section 5.1.1] From Fig. 6 a) and b) it comes across that the IWV retrieval is significantly improved by the addition of multiple input features. The highest accuracy is obtained with the full set of input features, and corresponds to a RMSE of 1.53 $kg\ m^{-2}$. On the other hand, including solely $T_B$ measurements in the input deteriorates the RMSE to nearly 6 $kg\ m^{-2}$. If only one input feature were available, all the versions would predict worse results than those given by reanalysis data. Including $T_B$ in the retrieval does not lead to the same leap in accuracy than for LWP (discussed in the following subsection), which was expected because the microwave frequency that is used here (89 GHz) is much less sensitive to water vapor. However, excluding $T_B$ from the input features degrades the RMSE to 2.56 $kg\ m^{-2}$, i.e. + 67 % error compared to the best version, which clearly shows that some information is extracted from the brightness temperature in the retrieval.

[Section 5.1.2] Figures 6 c) and d) show that for LWP retrieval, input features other than $T_B$ only bring second-order improvements, while they were shown to be crucial in the IWV retrieval. For instance, the addition of reanalysis data significantly improves the IWV retrieval, but only in a relatively minor way does it increase the accuracy of LWP retrieval. On the contrary, excluding $T_B$ from the input features leads to RMSE near 200 $g\ m^{-2}$ and $R^2 < 0.5$, i.e. to values with make the retrieval not relevant. This highlights once again that brightness temperature at 89 GHz is much more sensitive to liquid water than to water vapor. An additional reason for this high dependence on $T_B$ is that LWP at a given location can have a high temporal variability due to cloud dynamics in the atmospheric column, which might not always be captured in the time series of surface atmospheric variables, nor by ERA5 models which have a comparatively low spatial and temporal resolution.

**3  Anonymous referee #3**

*1.) It's very good and important to see, that an approach has been made to derive a globally valid retrieval algorithm for IWV and LWP observations from single channel microwave observations. This can be beneficial for different science applications, e.g. weather & climate but also in astronomy and radio propagation.*

*2.) The paper is well written and makes clear points. It nicely addresses the fact, that the combination of microwave radiometry in synergy with re-analysis output and standard environmental conditions can be advantageous.*

*3.) Although I strongly favor short and concise papers, the results presented here (especially in Sections 5 and 6) are – to an extent – kept rather minimal. The paper would benefit from a more detailed and quantitative discussion. See also my specific comments below.*

We thank the reviewer for their feedback. Following their comment, we have included a more detailed discussion on the algorithm's results in Sect. 5 and 6, which were largely rewritten.
* * *
1. *Lines 25-27: If the authors are referring to TBs in the microwave spectrum, please omit "radiative contribution of...aerosols"*

   We thank the referee for noting this mistake, which we removed.
* * *
2. *Lines 68 onward: I assume you performed a quality control for the radiosonde profiles used for retrieval development, if not please consider doing so. Depict checks concerning range (e.g. min/max) of atmospheric parameters, maximum ascent height, consistency checks concerning pressure and/or temperature gradient, etc...*

   Initially, no specific check was performed on the profiles, but unrealistic values were excluded at various stages (especially during the computation of LWP, IWV and TB with PAMTRA). However, this did not provide a full insight on the quality of the data, therefore a more thorough approach was implemented.

   A quality check was performed on each of the relevant variables (pressure, temperature, relative humidity), through the following steps: first, the minimum and maximum P (resp. T, RH) in a given range of altitudes were extracted from each radiosonde. When examining the distributions that are obtained, outliers were visible, which were then removed with a $10^{-4}$ quantile (upper and lower quantile). The atmospheric column was split into 9 ranges of altitudes, and this routine was performed for each. In total, 6395 profiles were flagged out and removed. It was ensured that this did not result in the systematic removal of some geographical locations. Following this step, the vertical profiles of pressure, temperature and relative humidity are used as input to the forward model, described in Sect. 3.
* * *
3. *Line 76: Please discuss what the relatively low vertical resolution could imply for the retrieval development. E.g., the coarse resolution of the relative humidity profile will influence where and*

*how many liquid layers are detected. What happens if this is significantly different for different radiosonde sites? This discussion could also be part of Section 3.1.*

This is a real concern that we encountered when designing the forward model. The crucial point was to ensure that the dataset that was used to train the algorithm was a realistic one, i.e. that it did not contain a statistical bias. We added a discussion in Sect. 3.1.

A further limitation of this cloud model is related to the relatively low resolution of the atmospheric profiles extracted from the radiosonde data (c.f. Sect. 2.1) that are used as an input. This might result in a misrepresentation of the cloud layers in their detection and their size. In order to ensure that this forward model resulted in the least possible bias, its results were compared against LWP values from ERA5 reanalysis data (Copernicus Climate Change Service, 2020). Even though the model might fail, on a given occurrence, to reproduce the actual liquid water profile in the atmospheric column, it should not produce a significant bias on average. This condition guarantees that the synthetic dataset that is used for training contains realistic – if not real – profiles, and this should therefore not degrade the quality of the retrieval algorithm. This cloud model was chosen over other commonly used ones (Decker model, Salonen model without correction, c.f. Mattioli 2009) for it was found to produce the least bias when compared to ERA5 LWP values (mean bias of 14 g m$^{-2}$ vs. 26 g m$^{-2}$ (resp. -24 g m$^{-2}$) for the unadjusted Salonen model (resp. the Decker model with 95 % threshold).
* * *
4. *Lines 92/93: Describe how far WProf and HATPRO were apart (in meters) and if you expect any corresponding uncertainties during retrieval application.*

This is a relevant information which was added to the text. We do not expect that it affects greatly the comparison, but there could be some edge cases where a cloud would be in the field of one radiometer and not the other.

The instruments were located approximately 65 m apart; this distance is small enough that it should in general not affect the comparison of the retrieved values from the two instruments. However, in some rare cases, it is possible that a cloud would overpass one of the radiometers, but not the other, leading to a discrepancy in the measured brightness temperatures.
* * *
5. *Section 3.2: I am missing a specification of the absorption models used for water vapor, oxygen and cloud liquid water. As previous studies have shown (e.g. Cimini et al. 2018, https://doi.org/10.5194/acp-18-15231-2018), this can be decisive for the absolute accuracy of your retrieval results. This aspect is nowhere discussed in the paper, should be, however.*

The description of the absorption model used was added to this section (Rosenkranz model for gases, which corrections from Turner 2009 and Liljegren 2005). It is the default one implemented in PAMTRA, and was used in some recent similar studies (e.g. Cadeddu 2017). As pointed out by the reviewer in this comment, there remains an irreducible uncertainty associated to those models, which the reader ought to have in mind.

Gaseous absorption is calculated using the default parameters in PAMTRA, i.e. with the model proposed by Rosenkranz 1998 and modifications from Turner 2009 and Liljegren 2005. It should be kept in mind that some irreducible uncertainty remains tied to the choice of these parameters in the radiative transfer model.
* * *
6. *Line 127: Please specify at which LWC (together with the assumed parameters of your gamma distribution, specify these as well), Mie effects become non-negligible and in how many cases in your data set this threshold is exceeded.*

Regarding the DSD, it was decided for the final version of the algorithm that a monodisperse distribution (as in Cadeddu 17) was more robust than the Gamma proposed by Karstens 1994, which introduced several empirical thresholds that could lead to artefacts and unrealistic discontinuities. This is also justified by the fact that, as long as the Rayleigh regime holds – i.e. in the range of validity of the algorithm – the choice of the DSD does not impact TB calculations. For the identification of potentially problematic profiles in terms of droplet size, criteria from Karstens 1994 were used and a discussion was included on this topic.

The cloud droplet size distribution (DSD) is chosen as a monodisperse distribution with radius $r_c = 20 \ \mu m$ following Cadeddu 2017, and scattering calculations are performed with Mie equations, assuming spherical particles.

In order to have a more rigorous grasp on when and how this drawback might affect the synthetic data set that is constructed – and hence the retrieval, criteria from Karstens 1994 were used. In their study, the authors distinguished three types of liquid water clouds based on the value of LWC at a given altitude; for each category of cloud, a different characteristic radius is chosen for the DSD. Assuming that Mie effects start to become an issue in the second category of clouds (*cumulus congestus*), identified for $LWC > 0.2g \ m^{-2}$, this corresponds on average to $LWP \geq 830 \ g \ m^{-2}$ and around 2% of the entire dataset fall into this category (i.e. the LWC threshold is exceeded in at least one altitude range). Taking the third category (*cumulonimbus*) with $LWC > 0.4 \ g \ m^{-2}$, this applies to 1% of the entire dataset and the threshold increases to $LWP > 1400 \ g \ m^{-2}$. Those values can serve as a benchmark to identify LWP values where Mie effects can typically contaminate the retrieval. However, edge cases can also exist where the total LWP is quite low, but a small layer of nearly-precipitating or drizzling cloud still contaminates the retrieval, without featuring extremely high total LWP.
* * *
7. *Section 4.1: I don't find any indication in the manuscript of how you are dealing with random uncertainty of your measurement variables, e.g. radiometric noise and T/q/p sensor uncertainty. If you want to simulate a realistic retrieval behavior, you need to put noise on your measurements (training, validation and test data set), otherwise you are assuming a "perfect relationship" between measurement and LWP/IWV, which you will never have in reality.*

This is a valid concern which we had neglected in the first place. In the latest version of the retrieval – which is presented in this revised version – a random Gaussian noise was added with a standard deviation of 0.5K as noted by Kuechler et al, 2017. This explanation was added to

section 4.1. Regarding the other surface parameters (temperature, relative humidity, pressure), we consider that it is not necessary to add noise, for the values used are already real measurements (i.e. from real sensors with noise).

In order to simulate realistic measurements, a random Gaussian noise was added to the modeled brightness temperatures, with a mean and standard deviation of resp. 0 K and 0.5 K; those values were identified by Kuechler et al. 2017 as the characteristics of the measurement noise of the 89-GHz radiometer.
* * *
8. *Line 140: Since the relationship between LWP/IWV and the TBs is, well I'd say, linear to moderately non-linear, I ask myself why you need to use $TB^4$? Can you quantify the retrieval improvement when using only TB and $TB^2$ in comparison to higher order terms?*

Following this comment and similar ones from the other referees, a more rigorous analysis was performed to determine the quantitative contribution of the higher-order terms. It was included in Sect. 5, and supporting figures are added in Appendix.

[Section 5.1.1] An analysis was conducted to identify the importance of higher order polynomials in the algorithm, a summary of which can be found in Appendix. It was found that the most accurate retrieval is obtained by including TB and $TB^2$. If higher order terms are added, this slightly reduces the accuracy of the retrieval, and also degrades its stability to TB miscalibration. On the other hand, including only TB, while it makes the algorithm slightly more stable, does not appear as the best solution for it has lower accuracy. Hence, the results which are presented here and in the further sections are those obtained using TB and $TB^2$.

[Section 5.1.2] The analysis of higher order terms' importance in the case of LWP retrieval shows that the best results are obtained by using TB polynomials up to the fourth order (cf. Appendix), while this does not affect significantly the stability of the retrieval to errors in TB. Let us highlight that in the case of a linear regression, one would expect the error to diverge when high-order polynomials are included. This is not the case here, because of the non-linear behavior of the neural network. Therefore, in the results which are shown here and further, "TB" implies that TB polynomials up to the fourth order are used.
* * *
9. *Begin with introducing your results in a general positive sense, make the reader feel like you are now going to present some great, interesting and relevant plots (which you mostly do!). I wouldn't begin Section 5.1 with two sentences that actually belong in the figure caption.*

An introductory sentence was added in the beginning of this section to open the discussion section in a more appealing way.

In this section, the algorithm is evaluated on the synthetic dataset (testing set), using different criteria; overall, results are encouraging and the retrieval appears to be robust. Some limitations can be identified, which will be discussed here. Additionally, an analysis on the impact of the
* * *
10. *Section 5.1: I assume Figure 5 illustrates the retrieval including all input parameters? Please state this clearly in the text and figure caption.*

    Indeed, Fig.5 illustrates the retrievals with all input parameters. This was clarified in the text and caption.

    [Text] The distribution of the error on the testing set is shown in Fig. 5, for the best version of the algorithm, which uses the full set of input features.

    [Figure caption] Results of the retrieval algorithms on the synthetic testing dataset. The best versions of the algorithms are presented, i.e. the ones which use the full set of input features.
* * *
11. *Lines 188 – 190: I think Figure 5 would benefit from two additional sub figures showing the bias as a function of binned IWV and LWP. Then you could quantify the statement you make in these two lines and elaborate a bit more on the bias behavior for smaller IWV.*

    As suggested by the reviewer, two subplots were added that illustrate the bias as a function of binned LWP and IWV. Please refer to the new figure (figure 5). The following sentence was added to the caption:

    Similarly, panels (e) and (f) show the distribution of mean bias across the range of IWV and LWP values.
* * *
12. *Lines 191 – 194: I'm a bit puzzled by Figure 6a. If you only use the 89 GHz TB then ERA5 performs significantly better. So, is there any sense of using this TB at all? I think you need to perform a retrieval without TB, just with all other parameters and add this one to your plots. This would help in putting the value of the 89 GHz TB in context. Then you need to discuss your results in more detail.*

    The IWV retrieval algorithm without TB is indeed an relevant point that was missing from the first version of the manuscript. We added it to the revised version, and it gives some valuable insight into the role of TB. A discussion on this point was included.

    From Fig. 6 a) and b) it comes across that the IWV retrieval is significantly improved by the addition of multiple input features. The highest accuracy is obtained with the full set of input features, and corresponds to a RMSE of 1.53 $kg\ m^{-2}$. On the other hand, including solely $T_B$ measurements in the input deteriorates the RMSE to nearly 6 $kg\ m^{-2}$. If only one input feature were available, all the versions would predict worse results than those given by reanalysis data. Including $T_B$ in the retrieval does not lead to the same leap in accuracy than for LWP (discussed

in the following subsection), which was expected because the microwave frequency that is used here (89 GHz) is much less sensitive to water vapor. However, excluding $T_B$ from the input features degrades the RMSE to 2.56 $kg\ m^{-2}$, i.e. $+$ 67 % error compared to the best version, which clearly shows that some information is extracted from the brightness temperature in the retrieval.
* * *
13. *Lines 196 and following: When you mention the RMSE of $86 gm^{-2}$, I assume you are applying the retrieval derived from the equally LWP-distributed training data set to the equally distributed test data set? I'm not sure.. please make clear.*

Indeed, the RMSE is calculated on the testing set which was preprocessed in the same way as the training set. This was not clear enough in the initial version of the text so the following modifications were added:

[Beginning of section 4.3] After preprocessing, LWP and IWV datasets were randomly split into training, validation and testing set (70 %-15 %-15 %) (...)

[Section 5.1.2] Let us underline that the subsampling which is performed on the dataset for the retrieval of LWP is applied to training, validation and testing sets: the results that are presented here are therefore computed on the testing set with a truncated distribution – i.e. after subsampling.
* * *
14. *Lines 199 – 201: You write: "with however a bias for low LWP values, which are slightly overestimated, and for large LWP ($> 800 gm^{-2}$) which are underestimated". Please apply my comment to Lines 188-190.*

As suggested by the reviewer, two subplots were added that illustrate the bias as a function of binned LWP and IWV. Please refer to the new figure (figure 5). The following sentence was added to the caption:

Similarly, panels (e) and (f) show the distribution of mean bias across the range of IWV and LWP values.
* * *
15. *Line 211: You write: "but only in a minor way does it increase the LWP retrieval's accuracy". But it does!? Going from noERA-noIWVpred-noGeo-noSurf to ERA-noIWVprednoGeo- noSurf reduces the RMSE from roughly 140 to 90 $gm^{-2}$. Or am I misinterpreting something wrong here?*

Fig. 6 was plotted again, with the most recent version of the algorithm including the modifications that were implemented following the reviewers' comments. The labels were made clearer. In the new figure, it is clearly visible that ERA5 input to the LWP retrieval does not bring a major improvement if it is used together with other input features. However, if considering only

TB+ERA as input features, this performs significantly better than TB alone.

Still, the accuracy of the algorithm drops severely when no other features are considered than brightness temperature (RMSE of 140 g m$^{-2}$). This means that, albeit second-order when taken individually, and somehow redundant when all used together, the secondary input features are efficient in incorporating statistical trends and climatological information to the retrieval during the training phase.
* * *
16. *Section 5.2: I like the idea of showing the sensitivity to instrument calibration offset, but only when I look at Figure 7, do I only see that you have looked at the effects for all different retrieval configurations and for continuously rising TB offset. Here again: please describe and discuss your results with more detail. To make your discussion complete, please add the "only-TB" retrieval to Figure 7a and 7b.*

Fig. 7 was updated and a little alleviated to make it more readable. The description and discussion of the figure was extended.

In order to assess the stability of the algorithm with respect to potential miscalibration or calibration drift of the radiometer, T$_B$ offsets were virtually added to the testing dataset before implementing the retrieval. Figure 7 illustrates the behavior of the algorithm when such a miscalibration, when a constant TB offset is present (varying from 0 to 5K). Panel a) shows that a 5 K offset in T$_B$ results in a 30 % increase in RMSE for the IWV estimations, which is non-negligible. Ensuring proper radiometer calibration thus seems crucial in constraining the error of this retrieval. For comparison, the 89-GHz radiometer presented in Kuechler et al, 2017 has a nominal accuracy of 0.5 K, after calibration. If the calibration cannot be ensured, and if there is no means to correct for miscalibration ( $> 3$K), it is preferable to use the algorithm that does not rely on TB, shown with the black dashed line.
* * *
17. *Section 5.3: Do you have any interpretation as of why the results over the Indian Peninsula are so much worse than elsewhere, even compared with sites at similar latitude? Is it possible that this is associated to the quality of the radiosondes or is there any other reason you can think of?*

18. *Line 242: Please describe how you think "humidity and temperature conditions" can lead to the discrepancy.*

We address those two comments jointly.
It is difficult to identify the most important factor that causes a high error in these areas. The underrepresentation of this range of latitudes and corresponding climates is most likely a critical point: unfortunately, there is to our knowledge little data available from these areas. The quality check that we performed on the radiosonde data did not suggest that there is a signficant and systematic difference in the quality of the data, although this cannot be fully excluded.

The temperature and humidity conditions, as well as the strong precipitation events that typically occur in those regions, are probably responsible for this discrepancy. Cases with high LWP

are more common under such climatic conditions, and it was observed in Sect. 4.1.2 that the accuracy of the algorithm decreases in that range. Tropical climates are underrepresented in the dataset, for little data is available from this region in comparison with mid-latitude areas: their specificity might therefore not be fully captured during the learning stage of the algorithm. This accounts at least partly for the enhanced error over the Indian peninsula and South-Eastern Asian islands.
* * *
19. *Figure 9: Can you explain the outliers (especially the vertical and horizontal "bar structures") in Figures 9a and 9b?*

We do not have clear-cut answer for this, but a few hypotheses that were added in the discussion (see below). Let us point out that the color scale is logarithmic and that these outliers are far less numerous than the bulk of the data, especially in the case of IWV.

Additionally, outliers are visible as vertical and horizontal bars close to the axes, for which two hypotheses are considered. One is that the distance between the two instruments was big enough, that in some cases a liquid water cloud would overpass one of the two instruments but not the other. Hence, HATPRO would measure a non-zero LWP while WProf would indicate a clear sky, or vice-versa. Also, measurement artefacts cannot be excluded, e.g. due the persistence of a liquid water film on the radome of either radiometer, after precipitation or due to condensation.
* * *
20. *I'm missing a discussion of Figure 11. One point would be, e.g. as also seen in Figure 6a, that the reanalysis performs better than the TB-only retrieval. Here, again it would help if you added a retrieval derived without any TB to discuss the overall TB value. Comparing such a retrieval against your ERA-Geo-Surf would tell you something about the impact of 89 GHz TB and if it's sensible to use it at all if you have the other parameters available.*

The retrieval without TB was included. Although brightness temperature does not have a similar impact as for LWP where it is the primary feature, it does improve the accuracy. It seems to have a similar weight as the other groups of features. In particular, it seems to increase the sensitivity to short temporal changes (visible in the comparison against HATPRO measurements). When averaging over 30 minutes, including TB does not result in significant improvement.

The top panels of Fig. 10 (which illustrates the error vs. HATPRO measurements) and Fig. 11 (error vs. value derived from radiosounding) shows that overall, the implementation of the different versions of the algorithm on the Payerne dataset matches the conclusions from the testing set results: more features lead to an enhanced precision of the retrieval. The accuracy drops when only one or two groups of input features are included, but no single group of features seem to increase the accuracy alone. There is however a difference between Fig. 10 b) and Fig. 11 b): in the latter, higher $R^2$ (and similar RMSE) is actually obtained from the algorithm that does not use $T_B$ in input, than with the full set of input features. This is at first surprising, but it was explained by taking a closer look at the results: the algorithm without $T_B$ leads to IWV values which are more smooth and less sensitive to short-time variations. These are not reflected in the comparison against radiosonde data, for which a 30-min averaging was implemented. When variations over a small timeframe are considered, the inclusion of $T_B$ improves the retrieval, as comes across from the comparison against HATPRO's measurements in Fig. 10.

21. *Line 261: Please quantify the "constant bias"*

The value of this bias (negative bias of $-1.8 kg\ m^{-2}$) was added, as well as a more quantitative description of the ERA5 bias over Payerne.

From Fig. 9 a) and c) it appears that the IWV retrieval has relatively limited spread but has a constant bias (-1.8 kg m$^{-2}$), which is visible both in the comparison against HATPRO (a) and radiosonde-derived measurements (c). This might be due to a bias in ERA5 data during this timeframe over the region (with a value of -4.1 kg m$^{-2}$), which is visible in ERA5 records during the entire campaign (not shown here) and for which there is no clear explanation at this stage.

22. *Lines 296 and following: In Fig 12, I'd also include results from a retrieval without TB. Another possibility for ERA5 outperforming the retrievals could maybe be fog? Could you include a discussion of the weather during ICE-POP?*

Figure 12 was also updated with the no-TB retrieval, which in the case of ICEPOP and the comparison against radiosonde measurement outperforms the other algorithms. A discussion on this is added to section 6.2.
Regarding weather conditions, a brief description is added to Sect. 2.2.3 and the topic is discussed further in Sect. 6.2.

[Section 2.2.3] A description of the data is presented in Gehring et al, 2020. During this campaign, the weather was generally cold and dry; nine precipitation events were recorded, and occasional fog was present (about 25 occurrences during the campaign timeframe).

[Section 6.2] Possibly, the dry and cold weather that was observed during the ICE-POP campaign featured little short-term variability and was associated with stable atmospheric conditions that were particularly well captured in ERA5 reanalyses. Snowfall events during the campaign, as well as occasional fog, can also bias the retrieval by enhancing brightness temperature.
The analysis of the ICE-POP data was taken a step further to explore the latter point. It appears that the IWV retrieval is most reliable in non-precipitating or cold conditions (...)

23. *Section 7: Can you elaborate a little on what the alternative would be to using the re-analysis? If, e.g. you would need quasi real-time retrieval results.*

A possibility would be to use, instead of reanalysis data, the output of forecast models (global models like IFS or GFS for instance, or regional models if available). They are likely to be reasonably accurate especially within a few hours of the model run. However, their performance might not be excellent in remote locations were few ground measurements and/or radisoundings are available. This drawback is also true for reanalysis data, although perhaps to a smaller extent.

Additionally, the retrieval that is presented here uses reanalysis data as an optional feature, that was shown to be very valuable. In the case where near real-time retrievals were necessary, the user could choose to use a version of the algorithm that does not rely on ERA5, but this would be detrimental, especially for the IWV retrieval. Another option would be to implement the algorithm with the output from forecast models (IWV and LWP) instead of reanalysis data. This approach was however not implemented as this stage.
* * *
24. *Figure 2: X-axis labelling of Figure 2b needs to read "LWP", not "IWV"*

We thank the referee for noting this mistake, which was corrected for.
* * *
25. *Figure 6: Do you need to make the bars orange with diagonal lines – just orange would probably make the text easier to read?*

The visual aspect of Fig. 6 was modified to make it more readable.
* * *
26. *Figure 9: It would help if you included in-plot statistics such as number of cases, bias, RMSE and $R^2$. Best do consistently with Figures 5 and 13.*

Following this suggestion, an annotation was added to the density plots and scatter plots (figures 5, 9 and 13) with statistical information: the number of plots, the root mean square error, as well as bias and correlation coefficient. The following sentence was added to the captions:

The size of the data set is indicated (N) as well as relevant error metrics (RMSE, mean bias, $R^2$).
* * *
27. *Figure 10: I think the ordering of the text in the bars is swapped, otherwise the plots make no sense to me.*

There was indeed a big mistake in the labels of the figure that was uploaded, the legend of the bars was swapped. This was corrected for.

---

## Author Response (AR3)

**Integrated water vapor and liquid water path retrieval using a single-channel radiometer**

**amt-2020-211**

**Responses to Reviewer 3**

Anne-Claire Billault-Roux and Alexis Berne

January 2021

We would like to thank Reviewer 3 for their additional comments, which were constructive and helpful for our analysis. In the present document, the comments of the referee are reported in italic, our responses in normal font and the corresponding modifications in the manuscript in blue.

**General**

*I like how the authors have improved the manuscript – specifically concerning a more detailed discussion of the results. I do have some remaining further general and specific points, which I would like the authors to bring forward more clearly.*
* * *
*For my point of view, the authors don't make it clear enough, why 89 GHz TB impacts the LWP retrieval more than IWV compared to the non-radiometric observations. To that extent it is important to note that LWP and atmospheric water vapor (mainly IWV and to lesser extent the profile shape) are basically the only contributors to the TB signal at 89 GHz. Since you can't retrieve two largely independent parameters (IWV and LWP) from only one measurement (89 GHz TB), you need to consider additional information that constrains the retrieval. The additional information the authors have (correctly) chosen (e.g. near-surface temperature and humidity, geographical location and altitude) is mostly correlated to IWV and hardly to LWP. This is the actual reason why LWP is more impacted by the 89 GHz TB: the additional information constrains the IWV and so the 89 GHz information can be used to retrieve the LWP. This needs to be made more clear. The authors, in contrast, also argument (see abstract, conclusions and within the main text) with a lower sensitivity of 89 GHz TB to IWV compared to LWP. This is a bit like comparing apples with oranges. If the authors would like to use this argument, they need to explicitly state the sensitivities (in TB/kgm-2) but these numbers then need to be related to the absolute variability of IWV, respectively LWP which again are not directly comparable.*

We thank the reviewer for this useful comment which brings more subtle insight into the problem. We reformulated some of our analysis to take this comment into account.

**Abstract**   While 89-GHz brightness temperature is crucial to LWP retrieval, only moderately does it contribute to IWV estimation, which is more constrained by the additional input features.

**Section 4.1**   Adding further information allows to disentangle IWV and LWP, which could not be achieved from the sole measurement of 89-GHz $T_B$. In this study, several categories of variables were included in the input features. The first category consists of $T_B$ and higher order polynomials (up to fourth degree) and is expected to have the greatest importance in the retrieval of LWP, while the other categories would likely be more correlated to IWV.

**Section 5.1**   This highlights that while environmental features are well correlated to IWV, they are not sufficient to provide a reasonable estimate of LWP, for which microwave radiometer measurements are critical.

**Section 6.2**   The algorithm largely relies on non-radiometric features, and this is even more the case in cold and dry environments like that of ICE-POP, where IWV is low.
* * *
*Error analysis: could the authors explain why they use R2 and not just R? R2 is often termed "explained variance". Also, I assume the RMSE is calculated in a way that includes the bias? Including information how RMSE is calculated would be great. I'm a bit puzzled about the "relative error" that is given for LWP and IWV. How was this calculated? E.g. the authors claim that the "relative error" is 7.2% for LWP values larger than 30 gm-2. But, if I look at Fig. 5, the RMSE is about 50 gm-2 at a target value of 100 gm-2 and about 100 gm-2 at a target value of 500 gm-2 which I would interpret as relative uncertainties of 50%, respectively 20%. Please clarify.*

We truly thank the reviewer for this relevant comment that allowed us to identify a flaw in our calculations of the relative error, which, indeed, were not coherent with the other error metrics shown.

- $R^2$ was replaced with R, for the sake of clarity, as suggested by the reviewer. This variable is simply used to illustrate the quality of the correlation between predicted and target values, which is not necessarily well captured by RMSE.

  Please see changes in Fig. 5, 6, 9, 10, 11, 12, 13.

- The equations for the error metrics are stated below (same for IWV). $LWP_{retrieved}$ and $LWP_{target}$ are length-N vectors with the values of predicted (i.e. algorithm-retrieved) and target LWP values, respectively. With to this definition, RMSE indeed include bias. Note that for the calculation of Relative error, $LWP_{target,k} = 0$ g/m$^2$ are excluded from the dataset, and that all LWP values are by definition positive. This information was included as a table in Appendix.

$$\text{RMSE} = \left[\frac{1}{N}\sum_{k=1}^{N}(\text{LWP}_{\text{retrieved,k}} - \text{LWP}_{\text{target,k}})^2\right]^{\frac{1}{2}}$$

$$\text{RelErr} = \frac{1}{N}\sum_{k=1}^{N}\frac{|\text{LWP}_{\text{retrieved,k}} - \text{LWP}_{\text{target,k}}|}{\text{LWP}_{\text{target,k}}}$$

$$\text{Bias} = \frac{1}{N}\sum_{k=1}^{N}(\text{LWP}_{\text{retrieved,k}} - \text{LWP}_{\text{target,k}})$$

  For reference, the definitions of the error metrics that are used in this section and further on are recalled in Appendix A1.

- As indicated before, there was a significant mistake in the values of relative error presented in the revised version of the manuscript. The values did not match the rest of the error metrics shown. This was corrected for. The correct values of the relative error on the test dataset, as calculated with the formula indicated above, are 29% and 18% when excluding LWP=0 and LWP>30 g m$^{-2}$, respectively. For IWV, the relative error is of 6.5 %.

**Abstract** The algorithm is shown to be quite robust although its accuracy is inevitably lower than that obtained with state-of-the-art multi-channel radiometers, with a relative error of 18 % for LWP (on cloudy cases with LWP $>$ 30 g m$^{-2}$) and 6.5 % on IWV.

**Section 5.1.1** The IWV retrieval algorithm yields a RMSE of 1.6 kg m$^{-2}$ on the testing set, which corresponds to a relative error of 6.5 %.

**Section 5.1.2** The LWP retrieval algorithm has a RMSE of 86 g m$^{-2}$ at best on the testing set (training set: 84 g m$^{-2}$ and validation set: 86 g m$^{-2}$). This corresponds to a relative error of 29 % on the testing set. Let us underline that the subsampling which is performed on the dataset for the retrieval of LWP is applied to training, validation and testing sets: the results that are presented here are therefore computed on the testing set with a truncated distribution – i.e. after subsampling. Additionally, if clear-sky cases are removed using 30 g m$^{-2}$ as a threshold value, following Loehnert and Crewell (2003), the relative error is 18 %.
* * *
*And last but not least: the presented IWV retrieval results should be briefly discussed in the context of ground-based GPS receivers, which are a world-wide standard methodology for deriving IWV.*

We included one sentence on this technology in the introductory section, and another one in the conclusion.

**Introduction** It should be noted that IWV retrievals with similar accuracy are obtained using GPS sensors, as first proposed by Bevis et al. (1994), but this widely used technique does however not allow for joint retrieval of LWP.

**Conclusion** If available through a separate sensor such as a GPS receiver, independent IWV measurements could be included in the algorithm which may lead to an enhanced precision of the LWP retrieval.

**Specific points**
* * *
*2.1 Radiosonde data set: please state how high a radiosonde ascent had to be in order to be used for NN training*

The information was added to this paragraph.

The vertical extent of the atmospheric profiles ranges from 1 to 50 km, with a 0.25 quantile of 11km, meaning the profiles largely cover the lower troposphere.

*3.1 Cloud liquid model: I see the ERA5 LWP comparison critical because, if I understand correctly, you assume the reanalysis LWP to be bias free. Could you comment on this in the text?*

This is a true limitation of this criterion, which should indeed be highlighted.

Inevitably, when using this criterion for the choice of the cloud liquid model, it is assumed that reanalysis values of LWP are themselves bias-free, which could be questioned, especially in extreme environments (e.g. Lenaerts et al. 2017).
* * *
*3.2 Radiative Transfer model: For completeness, please state which liquid water absorption model you have used. Also, I find the third paragraph difficult to follow and too lengthy. How can a LWC value correspond to a LWP value? This is dependent on cloud vertical extent. Can't you just justify a threshold in LWP above which with X% probability you'd expect significant precipitation leading to an additional drop size and DSD dependency of your TB?*

The information on the liquid water absorption model was included.

Liquid water absorption is modeled according to Ellison et al. (2007).

To our knowledge, there is no clear-cut relation between high LWP values and the occurrence of precipitation; indeed, an atmospheric profile featuring a cloud with a large vertical extent might have a high LWP, even though there is no precipitation. Besides, we have no direct way to identify precipitating cases in the radiosonde dataset. The method we proposed here was to use the cloud classification proposed by Karstens et al. (1994), to identify in our dataset the atmospheric profiles which might correspond to precipitating cases, or at least to clouds which diverge from Rayleigh DSDs. We tried to reformulate the paragraph to make this more clear. An LWC value as such does not correspond to an LWP value; the proposed method flags atmospheric profiles in which the LWC exceeds a threshold, and then the LWP of those profiles is calculated.

**There is no clear-cut relation between LWP values and the occurrence of precipitation, although the general trend is that higher LWP is related to more likely rain: as such, deviation from the Rayleigh regime is likely in high-LWP cases.** In order to have a more rigorous grasp on when and how this drawback might affect the retrieval, criteria from Karstens et al. (1994) were used. In their study, the authors distinguished three types of liquid water clouds based on the value of LWC at a given altitude; for each category of cloud, a different characteristic radius is chosen for the DSD. **Mie effects can start to become an issue in the second category of clouds (*cumulus congestus*), identified for LWC > 0.2 $\mathrm{g\,m^{-2}}$; in our dataset, the atmospheric profiles where this LWC theshold is exceeded (in at least one range gate) have, on average, a total LWP $\geq$ 830 $\mathrm{g\,m^{-2}}$, and around 2% of the entire dataset fall into this category**. Taking the third category (*cumulonimbus*) with LWC > 0.4 g m$^{-2}$, this applies to 1% of the entire dataset and **the average LWP threshold increases to 1400 g m$^{-2}$**. Those values can serve as a benchmark to identify LWP values where Mie effects can typically contaminate the retrieval. However, edge cases can also exist where the total LWP is quite low, but a small layer of nearly-precipitating or drizzling cloud still contaminates the retrieval, without featuring extremely high total LWP.
* * *
*5.1.2 LWP algorithm: the authors write: "Let us highlight that in the case of a linear regression one would expect the error to diverge when high-order polynomials are included. This is not the*

*case here because of the non-linear behavior of the neural network." I don't understand this argumentation. Doesn't the inclusion of higher order terms in a linear regression actually lead to a retrieval improvement if the original dependencies between observation and target variable are non-linear? And shouldn't the NN actually capture this non-linearity without having to include higher-order terms?*

- Including high-order polynomial terms in a linear regression is often beneficial, however it can lead to erroneous results outside of the training domain. For instance, if a linear algorithm were implemented on a case with high TB (higher than, say, the maximum TB considered in the training dataset), this could lead to a strong divergence. This would especially be an issue in the case of a miscalibrated instrument. The neural network somehow saturates this effect and prevents this excessive divergence.

- The NN could indeed capture the non-linearity without including polynomial terms in the input features (the results of the analysis show that including them brings some improvement, but not massive improvement). However, for it to fully capture the polynomial behavior without including those terms in the input features, it would require more NN parameters (hidden layers and/or neurons), which comes at the risk of overfitting. Since general knowledge of the "physics" of the problems indicates that the dependence of LWP on TB is approximately polynomial (e.g. Loehnert and Crewell 2003), it is reasonable to include those features in the input of the neural network.

To make this less confusing, we changed the word non-linear to saturating.
* * *
*5.2 Instrument calibration: I'm missing a discussion why in Fig. 7 there is practically no dependency on TB-offset (even if TB-offset = 5 K, corresponding to a delta LWP of say 250 gm-2? Please check...) for the TB-only retrieval, however very well on the TB + additional information retrievals. I'm pretty sure this has to do with the combined IWV and LWP dependency which I already commented on in the beginning.*

This is an interesting aspect of this Fig. 7, on which we propose a discussion:

It is noteworthy that for $T_B$-only retrievals, the addition of a $T_B$ offset does not result in a large increase of the error: for IWV, the addition of a 5K offset increases the RMSE from 5.6 to 6.2 kg m$^{-2}$; for LWP, the same offset leads to an increase from 139 to 142 g m$^{-2}$. This behavior is also observed when looking at how the bias, instead of the RMSE, increases with the addition of a $T_B$ offset (not shown). In both cases, the error increases more drastically when multiple features are included, than when only $T_B$ is used as input. One possible explanation for this effect is the following: when incorporating numerous input features, the algorithm is able to narrow down the range of possible IWV and LWP values in a given environmental context; in this constrained configuration, the correlation and sensitivity of the retrieval to $T_B$ are then enhanced, leading to a stronger influence of a $T_B$ offset.